# EFFICIENT NEURAL COMMON NEIGHBOR FOR TEMPORAL GRAPH LINK PREDICTION

## ABSTRACT

Temporal graphs are ubiquitous in real-world scenarios, such as social network, trade and transportation. Predicting dynamic links between nodes in a temporal graph is of vital importance. Traditional memory-based methods typically leverage the temporal neighborhood of interaction histories to generate node embeddings, which are then aggregated to predict links between source and target nodes. However, these methods primarily focus on learning individual node representations and often neglect the nature of pairwise representation learning aspect. While some recent methods attempt to capture pairwise features, they are less emphasized in large-scale datasets like TGB. Meanwhile, most of these existing methods tend to suffer from high computational complexity due to the repeated calculation of node embeddings. Motivated by the success of Neural Common Neighbor (NCN) for static graph link prediction, we propose **TNCN**, a temporal version of NCN for link prediction in temporal graphs. Based on a memory-based backbone instead of traditional static graph neural network, TNCN dynamically updates a temporal neighbor dictionary for each node, and utilizes multi-hop common neighbors between the source and target node to learn a more effective pairwise representation. We validate our model on five large-scale real-world datasets from the Temporal Graph Benchmark (TGB), and find that it achieves new state-of-the-art performance on three of them. Additionally, TNCN demonstrates excellent scalability on large datasets, outperforming popular GNN baselines by up to 6.4 times in speed.

## 1 INTRODUCTION

Temporal graphs are increasingly utilized in contemporary real-world applications. Complex systems such as social networks (Yang et al., 2017; Min et al., 2021; Nguyen et al., 2017), trade and transaction networks (Zhang et al., 2018; Yan et al., 2021), and recommendation systems (Wu et al., 2022; Yin et al., 2019) are prime examples. These systems evolve dynamically over time, exhibiting different characteristics. Recently, there has been a marked increase in the representation learning tasks on temporal graphs. At the same time, a prominent graph learning tool, Graph Neural Network (GNN) (Scarselli et al., 2008), has been developed to model node, link, and graph tasks. GNNs generally learn node embeddings by iteratively aggregating embeddings from neighboring nodes. They have demonstrated exceptional performance in numerous graph representation learning tasks.

There remains a significant gap between static graphs and the increasingly prevalent temporal graphs. Temporal graphs incorporate discrete or continuous timestamps attached to edges, providing a more precise depiction of the graph evolution process. Meanwhile, many methodologies for static graphs are not applicable due to the additional constraints imposed by timestamps on causality. One can only utilize nodes and edges that precede a given time, implicitly resulting in numerous graph instances to analyze. Building on the success of time sequence modeling, Kumar et al. (2019); Trivedi et al. (2019); Rossi et al. (2020) propose memory-based temporal graph networks aimed at learning both short- and long-term dependencies. These methods, particularly Temporal Graph Networks (TGN), have achieved notable success in temporal node classification. Additionally, Transformer-based models, such as those proposed by Wang et al. (2021a); Xu et al. (2020), employ the multi-head attention mechanism to capture both cross and internal relationships within a temporal graph.

While such memory- or attention-based methods are more emphasized and take more proportion in real-world datasets like TGB, these models may exhibit inherent flaws when addressing *link*

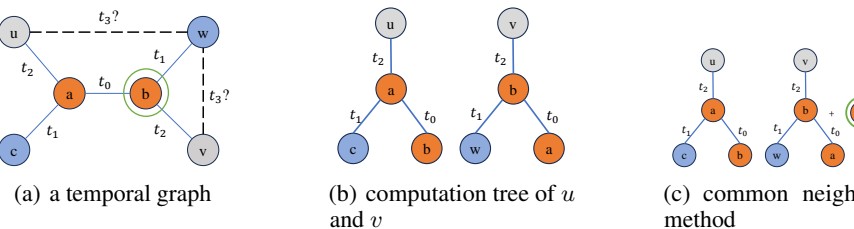

(a) a temporal graph  (b) computation tree of $u$ and $v$  (c) common neighbor method

Figure 1: Figure (a) shows a failure case of link prediction based on node-wise representation learning. Such methods cannot distinguish node $u$ and $v$ because they possess the same temporal computation tree in figure (b), thus generating the same node representation. However, when we try to learn their pair-wise representation, i.e. $(u, w)$ and $(v, w)$, we can observe that $v$ has a temporal common neighbor $b$ with node $w$ while $u$ doesn't, as shown in figure (c). Thus with the same computation graph, we only need to utilize the extra node $b$'s embedding to distinguish $(u, w)$ and $(v, w)$.

*prediction* tasks in graphs. Such tasks require the model to predict the existence of a target link. The aforementioned approaches primarily focus on node-wise representation learning, using the node embeddings of source and destination nodes to predict its existence. While straightforward, these methods often fail to precisely capture the node relationship and other complex structures. For example, in Figure 1, the node-wise method might struggle to predict whether node $w$ prefers to interact with node $u$ or $v$. Generally speaking, these models are constrained by their focus on node-level representation, limiting their ability to capture broader contexts and higher-order patterns.

Considering these shortcomings, Zhang & Chen (2018) highlight the importance of pair-wise representation and propose labeling trick to mark the source and destination nodes when learning their pair-wise representation by GNN. Labeling trick (Zhang et al., 2021) is shown to greatly enhance GNN performance on static graph link prediction. Building on this, Wang et al. (2021c); Luo & Li (2022); Yu et al. (2023) develop models that leverage information from temporal surrounding nodes, successfully extending the pair-wise learning approach to dynamic graphs. They extract features using decreasing-timestamp random walks or joint neighborhood structures to generate multi-level embeddings for the center nodes, facilitating node classification and link prediction with pair-wise representations. However, these graph-based models often incur significant computational and memory costs due to extracting temporal neighborhoods and applying message passing on them for each node/link to predict, hindering their application in real-world, large-scale scenarios.

In general, graph-based representation learning methods can significantly enhance model capabilities, yet their high computational cost limits widespread application. For example, the Temporal Graph Benchmark (TGB) (Huang et al., 2023) contains many high-quality real-world temporal graphs. However, most graph-based models choose to only evaluate on a subset of small graphs due to their unaffordable cost of training and evaluation on large-scale datasets. On the contrary, memory-based models (Rossi et al., 2020) update node representations sequentially following the event stream, thus is significantly faster than graph-based methods yet might lose important graph information. Motivated by these observations, we propose an expressive but efficient model, **Temporal Neural Common Neighbor** (**TNCN**), to combine the merits of both. By integrating a memory-based backbone with Neural Common Neighbor (Wang et al., 2023), TNCN learns expressive pairwise representations while maintaining high efficiency akin to memory-based models. Consequently, TNCN is suitable for large-scale temporal graph link prediction.

We conducted experiments on five large-scale real-world temporal graph datasets from TGB. TNCN achieved new SOTA results on 3 datasets and overall ranked first compared to 10 competitive baselines, demonstrating its effectiveness. To examine its scalability, we selected datasets with temporal edges ranging from $\mathcal{O}(10^5)$ to $\mathcal{O}(10^7)$ and node numbers ranging from thousands to millions. We found TNCN achieved 1.9x~4.7x speedup in training and 2.1x~6.4x speedup in inference compared to graph-based models, while maintaining a similar scale of time consumption as memory-based models.

## 2 RELATED WORK

### 2.1 MEMORY-BASED TEMPORAL GRAPH REPRESENTATION LEARNING

Temporal graph learning has garnered significant attention in recent years. A classic approach in this domain involves learning node memory using continuous events with non-decreasing timestamps.

Kumar et al. (2019) propose a coupled recurrent neural network model named JODIE that learns the embedding trajectories of users and items. Another contemporary work DyRep (Trivedi et al., 2019) aims to efficiently produce low-dimensional node embeddings to capture the communication and association in dynamic graphs. Rossi et al. (2020) introduces a memory-based temporal neural network known as TGN, which incorporates a memory module to store temporal node representations updated with messages generated from the given event stream. Apan (Wang et al., 2021b) advances the methodology by integrating asynchronous propagation techniques, markedly increasing the efficiency of handling large-scale graph queries. EDGE (Chen et al., 2021) emerges as a computational framework focusing on increasing the parallelizability by dividing some intermediate nodes in long streams each into two independent nodes while adding back their dependency by training loss. Chen et al. (2023) extend the update method for the node memory module, introducing an additional hidden state to record previous changes in neighbors. Complementing these efforts, additional contributions such as Edgebank (Poursafaei et al., 2022) and DistTGL (Zhou et al., 2023) have been directed towards formalizing and accelerating memory-based temporal graph learning methods.

## 2.2 Graph-based Temporal Graph Representation Learning

Subsequent works have incorporated the temporal neighborhood structure into temporal graph learning. CAWN (Wang et al., 2021c) employs random anonymous walks to model the neighborhood structure. TCL (Wang et al., 2021a) samples a temporal dependency interaction graph that contains a sequence of temporally cascaded chronological interactions. TGAT (Xu et al., 2020) considers the temporal neighborhood and feeds the features into a temporal graph attention layer utilizing a masked self-attention mechanism. NAT (Luo & Li, 2022) constructs a multi-hop neighboring node dictionary to extract joint neighborhood features and uses a recurrent neural network (RNN) to recursively update the central node's embedding. This information is then processed by a neural network-based encoder to predict the target link. DyGFormer (Yu et al., 2023), instead, leverages one-hop neighbor embeddings and the co-occurrence of neighbors to generate features, which are well-patched and subsequently fed into a Transformer (Vaswani et al., 2017) decoder to obtain the final prediction. LPFormer (Shomer et al., 2024) attempts to adaptively learn the pairwise encodings via graph attention module, utilizing relative position, ppr value and neighboring information to obtain the score. FreeDyG (Tian et al.) also utilizes historical interaction frequency akin to DyGFormer, afterwards transforming it with Fast Fourier Transform (FFT) and IFFT through the frequency domain. An MLP-mixer layer finally processes the output to generate the prediction. Another contemporary work CNE-N (Cheng et al., 2024) uses a hash table to map an interaction event to its position. It calculates the co-neighbor encoding for each (neighbor node - end node) pair within the local subgraph, recording the number of their common neighbors. These information are then concatenated to predict the probability of the future link.

## 2.3 Link Prediction Methods

Link prediction is a fundamental task in graph analysis, aiming to determine the likelihood of a connection between two nodes. Early investigations posited that nodes with greater similarity tend to be connected, which led to a series of heuristic algorithms such as Common Neighbors, Katz Index, and PageRank (Newman, 2001; Katz, 1953; Page et al., 1999). With the advent of GNNs, numerous methods have attempted to utilize vanilla GNNs for enhancing link prediction, revealing sub-optimal performance due to the inability to capture important pair-wise patterns such as common neighbors (Zhang & Chen, 2018; Zhang et al., 2021; Liang et al., 2022). Subsequent research has focused on infusing various forms of inductive biases to retrieve intricate pair-wise relationships. For instance, SEAL (Zhang & Chen, 2018), Neo-GNN (Yun et al., 2021), and NCN (Wang et al., 2023) have integrated neighbor-overlapping information into their design. BUDDY (Chamberlain et al., 2022) and NBFNet (Zhu et al., 2021) have concentrated on extracting higher-order structural information. Additionally, Mao et al. (2023); Li et al. (2024) have contributed to a more unified framework encompassing different heuristics.

## 3 Preliminaries

**Definition 3.1. (Temporal Graph)** Temporal graph can be typically categorized into two kinds, discrete-time (**DTDG**) and continuous-time (**CTDG**) dynamic graph. While DTDG can be repre-

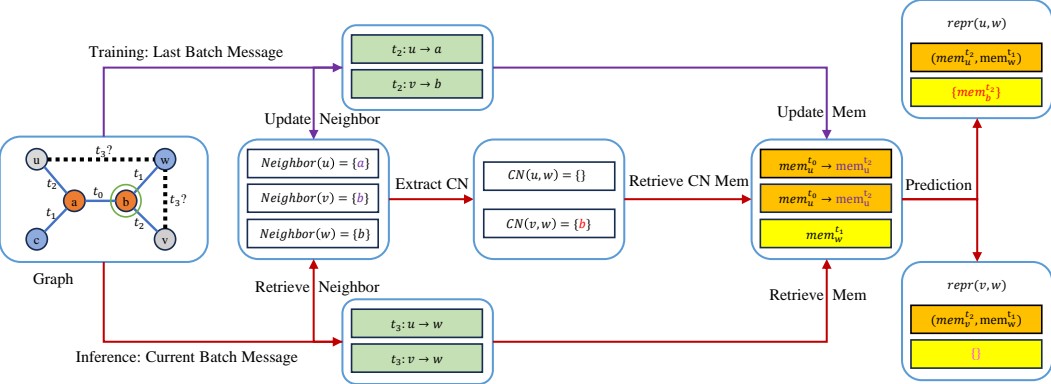

Figure 2: Pipeline of TNCN. TNCN operates through a sequential update and prediction framework that processes successive batches of messages. During the update phase, TNCN updates the neighbor dictionary and the node memory representations. In the prediction phase, the model retrieves neighbors to identify common neighbors, thereafter leveraging the representations of the target nodes and their common neighbors for prediction.

sented as a special sequence of graph snapshots in the form of $\mathcal{G} = \{(\mathcal{G}_1, t_1), (\mathcal{G}_2, t_2), \cdots, (\mathcal{G}_N, t_N)\}$, it can always be transformed into its corresponding form in CTDG. So here we mainly focus on continuous time temporal graph. We usually represent a CTDG as a sequence of interaction events: $\mathcal{G} = \{(u_1, v_1, t_1), (u_2, v_2, t_2), \cdots, (u_n, v_n, t_n)\}$, where $u, v$ stand for source and destination nodes and $\{t_i\}$ are chronologically non-decreasing. We use $\mathcal{V}$ to denote the entire node set, and $\mathcal{E}$ the entire edge set. Note that each node or edge can be attributed, that is, there may be node feature $x_u$ for $u$ or edge feature $e_{u,v}^t$ attached to the event $(u, v, t)$.

**Definition 3.2. (Problem Formulation)** Given the events before time $t^*$, i.e. $\{(u, v, t) \mid \forall t < t^*\}$, a link prediction task is to predict whether two specified node $u^*$ and $v^*$ are connected at time $t^*$.

**Definition 3.3. (Temporal Neighborhood)** Given the center node $u$, the $k$-hop temporal neighbor set $(k \geq 0)$ before time $t$ is defined as $N_k^t(u)$. A node $v$ is in $N_k^t(u)$ if there exists a $k$-length path between $u$ and $v$, i.e. $\exists(u, w_1, w_2, \cdots, w_{k-1}, v)$ where $w_i \neq w_j, \forall i \neq j$. We also define the $(\mathbf{i}, \mathbf{j})$**-hop common neighbor set** as follows: $w$ is an $(i, j)$-hop temporal common neighbor of $u$ and $v$ at time $t$ if $w \in N_i^t(u)$ and $w \in N_j^t(v)$. For simplicity we will denote the set as $\mathrm{CN}_{(i,j)}^t(u, v) = N_i^t(u) \cap N_j^t(v)$. Note that for $i = 0$ (or $j = 0$ similarly), we define the 0-hop temporal neighbor set as $N_0^t(u) = \{u\}$, and the $(0, j)$-hop common neighbor of $u$ and $v$ as $\mathrm{CN}_{(0,j)}^t(u, v) = N_0^t(u) \cap N_j^t(v) = \{u\} \cap N_j^t(v)$.

Finally, the $K$-hop temporal neighborhood of node $u$ at time $t$ is defined as: $\bigcup\limits_{k=0}^{K} N_k^t(u)$.

With $(i, j)$-hop neighborhood information, we can perceive the local structure to a large extent and distinguish the difference between multi-hop common neighbors more precisely.

## 4 METHODOLOGY

In this section, we introduce our **Temporal Neural Common Neighbor** (**TNCN**) model. TNCN comprises several key modules: the Memory Module, the Temporal CN Extractor, and the NCN-based Prediction Head. Special attention is given to the Temporal CN Extractor, designed to efficiently extract temporal neighboring structures and obtain multi-hop common neighbor information. The pipeline of TNCN is illustrated in Figure 2. And a pseudocode is also attached in Appendix E.

### 4.1 MEMORY MODULE

Different from the static graph neural network typically used in traditional NCN, our model TNCN adopts the **memory-based backbone** to efficiently store and update the node memory, eliminating the need for repeated computation of node embeddings within successive temporal batches.

The memory module stores node memory representations up to time $t$. When a new event occurs, the memory of the source and destination nodes is updated with the message produced by the event. The computation can generally be represented by the following formulas:

$$msg_{src}^t(u,v) = msgfunc_{src}(e_{u,v}^t), \qquad msg_{dst}^t(u,v) = msgfunc_{dst}(e_{u,v}^t).$$
$$mem_u^t = upd_{src}(mem_u^{t-}, msg_{src}^t(u,v)), \qquad mem_v^t = upd_{dst}(mem_v^{t-}, msg_{dst}^t(u,v)), \tag{1}$$

where $mem_u^{t-}$ stands for the embedding of node $u$ before time $t$. Here, *msgfunc* is a learnable function, such as a *linear projection* or a simple *identity function*. Note that for different edge directions, i.e., from source to destination and vice versa, the *msgfunc* and *updfunc* can be learned separately. The memory module aids the model in managing both long-term and short-term dependencies, thereby reducing the likelihood of forgetting. During training and inference, node memory evolves dynamically as events occur. Updates to this module reflect the dynamic nature of the temporal graph.

### 4.2 TEMPORAL CN EXTRACTOR

Our Temporal CN Extractor can efficiently perform multi-hop common neighbor extraction and aggregate their neural embeddings.

**Extended Common Neighbor.** The definition of multi-hop common neighbors (CN) is given in Definition 3.3, extending the traditional (1,1)-hop CN (i.e., nodes on 2-paths between $u$ and $v$) to arbitrary $(i,j)$-hop CN. Additionally, we define the zero-hop neighbor of a central node, i.e., $u$ is considered as a neighbor of itself, which will be utilized to calculate CNs with other nodes. Given source node $u$ and target node $v$, the (0,1)-hop and (1,0)-hop CN not only records the historical interactions between two nodes, but also reveals the frequency of their interactions.

**Efficient CN Extractor.** The CN Extractor is a crucial component of the TNCN model, contributing significantly to its high performance and scalability. It can efficiently gather pertinent information about a given center node and extract multi-hop common neighbors for a source-destination pair.

For each relevant node $u$, the extractor stores its historical interactions with other nodes as both source and destination. After a batch of events is processed by the model, the storage is updated with the latest interactions. This allows us to maintain a record of all historical interactions up to a certain timestamp, effectively constructing a dynamic lookup dictionary for fast retrieval during subsequent inference. To strike a balance between memory consumption and model capacity, we save only the most recent $K$ events and relevant nodes for each center node, where $K$ is a hyperparameter determined by the specific dataset.

To implement an efficient batch CN extractor, we organize the historical interactions in a *Sparse Tensor*, representing the temporal adjacency matrix. Then we perform **self-multiplication** to generate high-order adjacency connectivity. Sparse matrix **hadamard product** is finally employed to obtain separate $(i,j)$-hop CNs. All these operations can be efficiently implemented by sparse tensor operators and are supported by GPU to facilitate fast, batch processing. The detailed procedure can be found in Appendix F.

The utilization of **Multi-hop Common Neighbors** significantly boosts TNCN's performance, resulting in higher scores in temporal link prediction tasks. Furthermore, by employing sparse tensors, our model achieves substantial reductions in both storage requirements and computational complexity, thereby decreasing time consumption and enhancing efficiency. Here we also give a comparison between our TNCN and traditional NCN in Table 1.

### 4.3 NCN-BASED PREDICTION HEAD

For later link prediction or other downstream tasks, we first obtain the node embeddings from their memory:

$$emb_u^t = NN(mem_u^{t-}, \bigcup_{v \in N_1^t(u)} mem_v^{t-}, \bigcup_{t' < t} e_{u,v}^{t'}) \tag{2}$$

where NN has multiple choices, like Identity or simple static GNN (Bruna et al., 2013; Defferrard et al., 2016; Velickovic et al., 2017; Hamilton et al., 2017). In our implementation, we adopt Graph Transformer Convolution (Shi et al., 2020), which can pay more attention to the relation between different nodes and get local and global structure feature.

Table 1: The comparison of TNCN and NCN

|  | temporal scenario | backbone | arbitrary CN hops | batch-wise CN extraction |
|---|---|---|---|---|
| NCN | ✗ | traditional GNN | ✗ | ✗ |
| TNCN | ✔ | memory-based | ✔ | ✔ |

For source and destination nodes, we perform an element-wise product:

$$X_{u,v}^t = emb_u^t \otimes emb_v^t. \tag{3}$$

For multi-hop CN nodes, we aggregate their embeddings in each hop with sum pooling:

$$NCN_{(i,j)}(u,v) = \underset{w \in \text{CN}_{(i,j)}^t(u,v)}{\oplus} emb_w^t. \tag{4}$$

These embeddings are then concatenated as the final pair-wise representation:

$$repr(u,v) = [X_{u,v}^t \;||\; (\overset{K}{\underset{i,j}{||}}) NCN_{(i,j)}(u,v)]. \tag{5}$$

In the above, $\otimes$, $\oplus$, and $||$ stand for element-wise product, element-wise summation, and concatenation of vectors, respectively. The pair-wise representation $repr(u,v)$ for nodes $u$ and $v$ is fed to a projection head to output the final link prediction.

## 5 EFFICIENCY AND EFFECTIVENESS OF TNCN

In this section, we explore the two principal benefits of TNCN: efficiency and effectiveness. These advantages are demonstrated through an analysis of two core components within the framework for temporal graph link prediction: graph representation learning and link prediction methods. We categorize graph representation learning modules into two types: memory-based and $k$-hop-subgraph-based, according to **the temporal scope of evolved events**. Memory-based modules exhibit superior time complexity while maintaining good expressiveness in some situations, striking a balance between efficiency and performance. Furthermore, we highlight the deficiencies of existing link prediction methods on temporal graph learning and introduce the extended common neighbor approach. This method serves as a complementary addition for learning pair-wise representations while eliminating the necessity for message passing on entire graphs. Both our graph representation learning and link prediction techniques are designed with a unified optimization objective: **to avoid message passing on entire subgraphs in favor of non-repetitive operations**, culminating in a cohesive solution that is both efficient and effective.

### 5.1 GRAPH REPRESENTATION LEARNING

Graph representation learning aims to develop an embedding function, denoted as $Emb$, which learns an embedding for each node encoding its structural and feature information within the graph. Specifically, given a new event represented as $(u, v, t)$, the function $Emb$ leverages prior events to generate meaningful embeddings. Approaches in this domain diverge in their handling of temporal dynamics; some opt to maintain a dynamic embedding for each node that is incrementally updated with each new event, while others choose to recalculate node embeddings by considering the entire historical context of events, thereby providing a more comprehensive reflection of past interactions. We classify these methodologies into two distinct types based on their operational mechanisms.

**Definition 5.1. Memory-based approach.** Given a new event $(u, v, t)$, if $Emb$ conforms to the following form, the method is referred to as a memory-based approach.

$$Emb(u,t) = f_{emb}(Mem(u,t')), \quad Mem(u,t) = f_{mem}(Mem(u,t'), Mem(v,t'), e_{u,v}^t, t - t'),$$
$$Emb(v,t) = f_{emb}(Mem(v,t')), \quad Mem(v,t) = f_{mem}(Mem(v,t'), Mem(u,t'), e_{v,u}^t, t - t'), \tag{6}$$

where $f_{emb}$ and $f_{mem}$ are two learnable functions.

**Definition 5.2.** $k$-**hop-subgraph-based approach.** Given a new event $(u, v, t)$, if $Emb$ conforms to the following form, the method is defined as a subgraph-based approach.

$$Emb(u, t) = f_{emb}(\mathcal{G}_{u,<t}^k), \quad Emb(v, t) = f_{emb}(\mathcal{G}_{v,<t}^k), \tag{7}$$

where $\mathcal{G}_{u,<t}^k$ is a subgraph induced from $\mathcal{G}$ by node $u$'s $k$-hop temporal neighborhood $\overset{K}{\underset{k=0}{\cup}} N_k^t(u)$, containing only the edges (events) with time $t' < t$, and $f_{emb}$ is a learnable function.

**Effectiveness.** The analysis begins by assessing the effectiveness of the two paradigms. To do so, we first introduce the concept of $k$-hop event (LOV´ASZ et al., 1993).

**Definition 5.3.** ($k$-hop event & monotone $k$-hop event) A *$k$-hop event* is a sequence of consecutive edges $\{(u_i, u_{i+1}, t_{u_i,u_{i+1}}) \mid i \in \{0, \dots, k-1\}, k \geq 1\}$ connecting the initial node $u_0$ to the final node $u_k$. For example, $\{(u, x, t'), (x, v, t)\}$ is a 2-hop event. In the case where $k = 1$, the $k$-hop event reduces to a single interaction $(u, v, t)$. A *monotone $k$-hop event* is a $k$-hop event in which the sequence of timestamps $\{t_{u_i,u_{i+1}} \mid i \in \{0, \dots, k-1\}, k \geq 1\}$ is strictly monotonically increasing.

Then, we analyze the expressiveness of the two approaches in terms of encoding $k$-hop event.

**Theorem 5.4.** *(Ability to encode $k$-hop events). Given a $k$-hop event $\{(u_i, u_{i+1}, t_{u_i,u_{i+1}}) \mid i \in \{0, \dots, k-1\}, k \geq 1\}$, if the node embedding of $u_0$ at time $t_{u_0,u_1}$ can be reversely recovered from the encoding $Enc(\{(u_i, u_{i+1}, t_{u_i,u_{i+1}}) \mid i \in \{0, \dots, k-1\}, k \geq 1\})$, then we say the encoding function Enc is capable of encoding the $k$-hop event. The following results outline the encoding capabilities of different learning paradigms:*

- *Memory-based approaches can encode any $k$-hop events with $k = 1$.*

- *Memory-based approaches can encode any monotone $k$-hop events with arbitrary $k$.*

- *$k$-hop-subgraph-based approaches can encode any $k'$-hop events with $k' \leq k$.*

From Theorem 5.4, we can conclude that 1) memory-based approaches have superior expressiveness in encoding $k$-hop events compared to 1-hop-subgraph-based approaches, 2) memory-based approaches have superior expressiveness in encoding monotone $k$-hop events than $k'$-hop-subgraph-based approaches when $k' < k$, and 3) $k$-hop-subgraph-based approaches are not less expressive than memory-based approaches when $k$ is large enough.

While the memory-based approach does not consistently rival the expressiveness of the $k$-hop-subgraph paradigm, it possesses advantages in monotone events and long-history scenarios (where $k$-hop subgraphs would be unaffordable to extract).

**Corollary 5.5.** *If we use up to $k$ hop neighborhood information of central node $u$, then TNCN can capture at least $(k + 1)$-hop subgraph information around $u$.*

This is because TNCN with memory-based backbone can obtain additional 1-hop information regardless of the time monotony, i.e. arbitrary central node can interact with any neighbor when the edge between them exists. This property can extend TNCN's capability for free.

**Efficiency.** We then turn our attention to the efficiency of the two approaches. A pivotal factor is the frequency with which individual events are incorporated into computations. In memory-based approaches, each event is utilized **a single time** for learning, immediately following its associated prediction. Conversely, in the $k$-hop-subgraph-based method, an event may be employed **multiple times**, as it is revisited in different nodes' temporal neighborhood and repeated been processed within each subgraph's encoding (such as message passing) process. This discrepancy leads to divergent cumulative frequencies of event utilization throughout the learning process and results in the huge efficiency advantage of memory-based methods. We formalize this observation in the following:

**Theorem 5.6.** *(Learning method time complexity). Denote the time complexity of a learning method as a function of the total number of events processed during training. For a given graph $\mathcal{G}$ with the number of nodes designated as $|\mathcal{N}|$ and the number of edges as $|\mathcal{E}|$, the following assertions hold:*

- *For memory-based approaches, the time complexity is $\Theta(|\mathcal{E}|)$.*

- *For $k$-hop-subgraph-based approaches with $k = 1$, the lower-bound time complexity is $\Omega\left(\frac{|\mathcal{E}|^2}{|\mathcal{N}|}\right)$, and the upper-bound time complexity is $\mathcal{O}\left(\frac{|\mathcal{E}|^2}{|\mathcal{N}|} + |\mathcal{E}||\mathcal{N}|\right)$.*

- *For $k$-hop-subgraph-based approaches with $k = 2$, the upper-bound time complexity is* $\mathcal{O}\left(\left(\frac{|\mathcal{E}|^2}{|\mathcal{N}|} + |\mathcal{E}||\mathcal{N}|\right)^{\frac{3}{2}}\right)$.

The proof is attached in Appendix H with part of the proof based on a classic conclusion from de Caen (1998) in the graph theory. Following Theorem 5.6, it becomes evident that the computational overhead incurred by a memory-based method is significantly lower than that of a $k$-hop-subgraph-based method, particularly as the value of $k$ increases. These results highlight the advantages of memory-based methods in mitigating the computational efficiency challenges associated with large-scale temporal graphs.

## 5.2 Link Prediction Technique

With the node embeddings obtained from the graph representation learning step, link prediction techniques involve aggregating the node embeddings in some way into link representations for link prediction. Most previous methods simply concatenate the source and destination node embeddings as the link representation $Emb(u, t) \,||\, Emb(v, t)$, losing a great amount of pairwise features (as illustrated in Figure 1). In order to use labeling trick, an alternative approach involves extracting a separate subgraph for each link to predict, which leads to good results but also significantly increases the complexity. TNCN, on the contrary, utilizes neural common neighbors as a *decoupled and flexible* solution. This approach can be *seamlessly integrated into existing memory-based methods* with minimal computational overhead, while retaining important structural information for link prediction.

**Effectiveness.** In the following, we first demonstrate the effectiveness of TNCN by showing that it can capture three important pairwise features commonly used as effective link prediction heuristics, namely Common Neighbors (CN), Resource Allocation (RA), and Adamic-Adar (AA) (Newman, 2001; Adamic & Adar, 2003; Zhou et al., 2009), borrowing from NCN (Wang et al., 2023).

**Theorem 5.7.** *TNCN is strictly more expressive than CN, RA, and AA.*

Essentially, TNCN uses node memory to substitute the constant or degree-based values in the three features, and the memory update scheme is sufficient to learn such values. In comparison, an approach that merely concatenates node-wise representations proves inadequate in capturing these heuristics.

**Efficiency.** We then address the efficiency of TNCN. The computation of TNCN can be divided into two primary components: the generation of neighbor nodes' embeddings and the execution of common neighbor lookups. Concerning the former, the memory-based approach intrinsically tracks all node embeddings, thus obviating the need for re-computation. As for the latter, we implement a fast common neighbor search algorithm by leveraging a sparse matrix structure that supports batch operations. Collectively, these factors contribute to the minimal additional overhead of TNCN compared to memory-based methods.

## 6 Experiments

This section assesses TNCN's effectiveness and efficiency by answering the following questions:

**Q1:** What is the performance of TNCN compared with state-of-the-art baselines?
**Q2:** What is the computational efficiency of TNCN in terms of time consumption?
**Q3:** Do the extended common neighbors bring benefits to original common neighbors?

### 6.1 Experimental Settings

**Datasets** We evaluate our model on five large-scale real-world datasets for temporal link prediction from the ***Temporal Graph Benchmark*** (Huang et al., 2023). These datasets span several distinct fields: co-editing network on Wikipedia, Amazon product review network, cryptocurrency transactions, directed reply network of Reddit, and crowdsourced international flight network. They vary in scales and time spans. Additional details about the datasets are provided in Appendix A. We set the evaluation metric as **Mean Reciprocal Rank (MRR)** consistent with the TGB official leaderboard.

**Baselines** We systematically evaluate our proposed model against a diverse set of baselines known for their strong capacity to represent temporal graph dynamics. These include: a heuristic algorithm

Table 2: Test Performance of different models under MRR metric. The top three are emphasized by red, blue and **bold** fonts. 'NA' denotes scenarios where a specific method was either not applied to the dataset or was unable to complete the validation and testing phases within a reasonable timeframe.

| Model | Wiki | Review | Coin | Comment | Flight |
|---|---|---|---|---|---|
| TGN | 0.528±0.058 | 0.387±0.021 | 0.737±0.031 | **0.622±0.023** | **0.705±0.018** |
| DyGformer | 0.798±0.010 | 0.224±0.015 | **0.752±0.004** | 0.670±0.001 | NA |
| NAT | **0.749±0.010** | 0.341±0.020 | NA | NA | NA |
| CAWN | 0.711±0.006 | 0.193±0.001 | NA | NA | NA |
| Graphmixer | 0.118±0.002 | 0.521±0.015 | NA | NA | NA |
| TGAT | 0.141±0.007 | 0.355±0.012 | NA | NA | NA |
| TCL | 0.207±0.025 | 0.193±0.009 | NA | NA | NA |
| DyRep | 0.050±0.017 | 0.220±0.030 | 0.452±0.046 | 0.289±0.033 | 0.556±0.014 |
| Edgebank(tw) | 0.571 | 0.025 | 0.580 | 0.149 | 0.387 |
| Edgebank(un) | 0.495 | 0.023 | 0.359 | 0.129 | 0.167 |
| TNCN-official | 0.724±0.001 | **0.419±0.009** | 0.770±0.006 | 0.727±0.012 | 0.817±0.004 |
| TNCN-ns | 0.778±0.001 | 0.427±0.006 | 0.771±0.004 | 0.596±0.008 | 0.831±0.003 |

Edgebank (Yu et al., 2023), memory-based models DyRep (Trivedi et al., 2019) and TGN (Rossi et al., 2020) that obviate the need for frequent temporal subgraph sampling, and GraphMixer (Cong et al., 2023) which primarily employs an MLP-mixer. Additionally, we include various graph-based models such as CAWN (Wang et al., 2021c), NAT (Luo & Li, 2022), DyGFormer (Yu et al., 2023), TGAT (Xu et al., 2020), and TCL (Wang et al., 2021a), which learn from neighborhood structure information.

Here we evaluate our TNCN under two similar but different settings, the official setting ("official") and the new setting ("ns"). "TNCN-official" strictly complies to the **official setting** of TGB evaluation policy, using both *streaming setting* and *lag-one scheme* for both memory update and neighborhood awareness. Streaming setting means the information of the validation and test sets can only be employed for updating the memory without any back propagation. Lag-one scheme implies that the model can access only the information from **before the current batch** for predictions; in other words, the latest usable batch is the previous one. This applies to not only the memory, but also the **neighborhood awareness**. "TNCN-ns" obeys the streaming setting but considers the interactions within the same batch before the current prediction time. This allows the model to utilize more recent neighborhood information, potentially giving it unfair advantages in datasets where recent interactions are crucial. Methods "DyGFormer", "NAT" and "Graphmixer" reported on TGB leaderboard use the latter setting while others use the former. For a fair evaluation and comparison, here we display the performance of our TNCN under both settings.

## 6.2 EXPERIMENTAL RESULTS

**Reply to Q1: TNCN possesses remarkable performance.** We conducted comprehensive evaluations of prevailing methods on the TGB. The main results are summarized in Table 2. It is evident from the table that TNCN attains new SOTA performance on three out of five datasets. Additionally, TNCN demonstrates competitive results on the remaining two datasets. TNCN almost consistently surpasses memory-based models such as TGN and DyRep, which can be attributed to its utilization of supplementary structural information. Even in comparison to powerful graph-based models, including NAT and DyGFormer, TNCN still matches or exceeds their performance, underscoring the effectiveness of its integration of node memory and pair-wise features. The only dataset where TNCN still exhibits a large performance gap from the best baseline is the Review dataset. This may be ascribed to the dataset's predominant reliance on edge feature embedding and its high "surprise" index, wherein prior events have a diminished correlation with subsequent events, potentially reducing the impact of CNs. (*Surprise index* computes the ratio of test edges that are not seen during training, i.e., $\frac{|E_{test}/E_{train}|}{|E_{test}|}$. More details can be referred to Appendix A.)

We have also conducted some additional experiments such as transductive and inductive settings, TGN with heuristics, *etc.*, uncovering strong performance of our model TNCN. More results can be referred to Appendix B.

**Reply to Q2: TNCN shows great scalability on large datasets.** To evaluate computational efficiency,

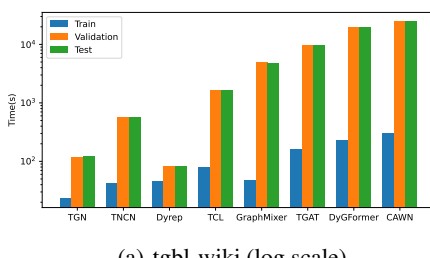

(a) tgbl-wiki (log scale)

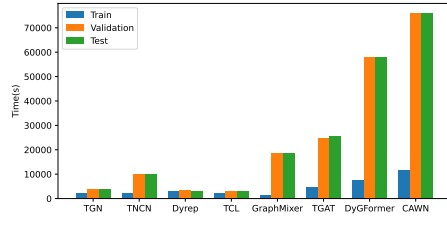

(b) tgbl-review (linear scale)

Figure 3: Time Consumption of Memory and Graph-based Method on Wiki and Review Datasets

Table 3: Test performance of TGN and TNCN with different ranges of common neighbors

| Model | Wiki | Review | Coin | Comment |
|---|---|---|---|---|
| TGN | 0.528 | 0.387 | 0.737 | 0.622 |
| TNCN-1-hop-CN | 0.621 | 0.419 | 0.737 | 0.641 |
| TNCN-0∼1-hop-CN | 0.720 | 0.298 | 0.739 | 0.727 |
| TNCN-0∼2-hop-CN | 0.724 | 0.317 | 0.770 | 0.662 |

we collected the time consumption when applying all baselines to the Wiki and Review datasets, as depicted in Figure 3. Compared with memory-based methods, TNCN exhibits a comparable order of magnitude in terms of time consumption. However, when benchmarked against graph-based models, TNCN demonstrates a substantial acceleration, achieving approximately 1.9 to 4.7 times speedup during the training phase and a 2.1 to 6.4 times increase in inference speed. Notably, the scalability concerns become even more evident as the size of the dataset expands; several graph-based models cannot complete the validation and testing processes within a reasonable time budget. The primary factors contributing to TNCN's efficiency are the synergistic, time-efficient design of its two core components and the implementation of the Efficient CN Extractor that facilitates batch operations through parallel processing. More detailed statistics can be referred to Appendix D.

**Reply to Q3: Extended CN brings improvements.** To elucidate the benefits of extended CN, we conducted an ablation study under **official setting** on the hop range of common neighbors. The results are shown in Table 3. Here we use notation "$k$-hop CN" to simply denote the CNs up to $(k, k)$-hop. The conventional NCN method considers only (1,1)-hop CN. However, this approach may not be universally applicable across all temporal networks. For instance, *bipartite graphs* lack such (1,1)-hop CN in their structure, necessitating the consideration of 2-hop CN. Additionally, memory-based methods may omit a notable aspect: they generally find it difficult to quantify the frequency of interactions between a given pair of nodes, which brings the need for 0-hop neighborhood.

To address these limitations, we have expanded the original (1,1)-hop CN to 0∼k-hop CN. Table 3 presents the model's performance under different hops of CN. The results indicate that TNCN utilizing 0∼1-hop CN markedly surpasses the (1,1)-hop CN on various datasets. This enhancement underscores the significance of the introduced 0-hop neighbors to our architecture. Nevertheless, the inclusion of 2-hop CN yields mixed results across datasets.

# 7 CONCLUSION AND LIMITATION

We propose TNCN for temporal graph link prediction, which employs a temporal common neighbor extractor combined with a memory-based node representation learning module. TNCN has achieved new state-of-the-art results on several real-world datasets while maintaining excellent scalability to handle large-scale temporal graphs.

However, based on our observation of TNCN's performance on the TGBL-Review dataset, there are some limitations in our model. Specifically, datasets with high surprise values, such as the Review dataset, tend to make it more challenging for TNCN to accurately predict the probability of future connections. This indicates that while TNCN performs well overall, it may struggle with datasets that exhibit high variability or unexpected patterns. Further research is needed to address these challenges and improve the model's robustness in such scenarios.

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

# A  DATASETS

Table 4 shows some detailed datasets statistics of TGB and 5 shows several temporal graph datasets commonly used by previous work. Through the two tables we can observe that TGB official datasets possess temporal graphs with larger scale to 10 million, 10 times surpassing the largest previous datasets such as LastFM, Flight, Contact. With the aim to examine our TNCN model's efficiency, we choose the increasingly accepted datasets TGB in the main table.

Table 4: TGB Dataset Statistics

| Dataset | Domain | Nodes | Edges | Steps | Surprise | Edge Properties |
|---------|--------|-------|-------|-------|----------|-----------------|
| tgbl-wiki | interact | 9,227 | 157,474 | 152,757 | 0.108 | W: ×, Di: ✓, A: ✓ |
| tgbl-review | rating | 352,637 | 4,873,540 | 6,865 | 0.987 | W: ✓, Di: ✓, A: × |
| tgbl-coin | transact | 638,486 | 22,809,486 | 1,295,720 | 0.120 | W: ✓, Di: ✓, A: × |
| tgbl-comment | social | 994,790 | 44,314,507 | 30,998,030 | 0.823 | W: ✓, Di: ✓, A: ✓ |
| tgbl-flight | traffic | 18143 | 67,169,570 | 1,385 | 0.024 | W: ×, Di: ✓, A: ✓ |

Here "Surprise" (Poursafaei et al., 2022) refers to the ratio of test edges that are not seen during training, which can be calculated as $\frac{|E_{test}/E_{train}|}{|E_{test}|}$. Low surprise index implies that memory-based methods such as Edgebank (Poursafaei et al., 2022) may potentially achieve good performance, while high surprise may require more inductive capability. The surprise index varies across TGB datasets.

Table 5: Previous Dataset Statistics

| Datasets | Domains | Nodes | Links | N&L Feat | Bipartite | Duration | Unique Steps | Time Granularity |
|----------|---------|-------|-------|----------|-----------|----------|--------------|------------------|
| Wikipedia | Social | 9,227 | 157,474 | – & 172 | ✓ | 1 month | 152,757 | Unix timestamps |
| Reddit | Social | 10,984 | 672,447 | – & 172 | ✓ | 1 month | 669,065 | Unix timestamps |
| MOOC | Interaction | 7,144 | 411,749 | – & 4 | ✓ | 17 months | 345,600 | Unix timestamps |
| LastFM | Interaction | 1,980 | 1,293,103 | – & – | ✓ | 1 month | 1,283,614 | Unix timestamps |
| Enron | Social | 184 | 125,235 | – & – | × | 3 years | 22,632 | Unix timestamps |
| UCI | Social | 1,899 | 59,835 | – & – | × | 196 days | 58,911 | Unix timestamps |
| Flights | Transport | 13,169 | 1,927,145 | – & 1 | × | 4 months | 122 | days |
| UN Trade | Economics | 255 | 507,497 | – & 1 | × | 32 years | 32 | years |
| Contact | Proximity | 692 | 2,426,279 | – & 1 | × | 1 month | 8,064 | 5 minutes |

# B  ADDITIONAL EXPERIMENTAL RESULTS

## B.1  TRANSDUCTIVE AND INDUCTIVE EXPERIMENTS ON PREVIOUSLY SMALL AND MEDIUM DATASETS

Table 6 shows the performance of different models on some small and medium datasets previously used in dynamic graph link prediction.

## B.2  COMPARISON WITH SOME CLASSIC HEURISTIC METHODS

Table 7 exhibits the result between TGN with some classic heuristics and TNCN under official setting on tgbl-wiki dataset. Here heuristics consist of CN (Barabási & Albert, 1999), RA (Zhou et al., 2009), AA (Adamic & Adar, 2003), PPR (Page et al., 1999; Jeh & Widom, 2003) and ELPH (Chamberlain et al., 2022). In these heuristic methods, the heuristic statistics are concatenated with TGN embedding to obtain final predictions. From the table we can see that these basic heuristics such as CN and RA do not bring performance improvement. However, some sophisticated heuristics like graph sketching in ELPH can enhance the backbone's capability. Nevertheless, using these heuristics cannot outperform a more generalized model like our TNCN, which comprehensively takes neighborhood nodes' representations into account.

Table 6: Average Precision (AP) under Transductive and Inductive settings on small and medium dataset. The best is in **bold** font.

| Method | Wikipedia | Reddit | Mooc | Lastfm |
|---|---|---|---|---|
| Transductive | | | | |
| CAWN | 98.62±0.05 | 98.66±0.09 | 80.15±0.25 | 86.99±0.06 |
| JODIE | 96.15±0.36 | 97.20±0.05 | 80.23±2.44 | 70.85±2.13 |
| DyRep | 95.81±0.15 | 98.00±0.19 | 81.97±0.49 | 71.92±2.21 |
| TGAT | 96.94±0.06 | 98.52±0.02 | 85.84±0.15 | 73.42±0.21 |
| NAT | 98.68±0.04 | 99.10±0.09 | 86.54±0.02 | 88.56±0.02 |
| TCL | 96.47±0.16 | 97.53±0.02 | 82.38±0.24 | 67.27±2.16 |
| DyGFormer | 99.03±0.02 | 99.22±0.01 | 87.52±0.49 | **93.00±0.12** |
| FreeDyG | **99.26±0.01** | **99.48±0.01** | 89.61±0.19 | 92.15±0.16 |
| EdgeBank | 90.37±0.00 | 94.86±0.00 | 57.97±0.00 | 79.29±0.00 |
| GraphMixer | 97.25±0.03 | 97.31±0.01 | 82.78±0.15 | 75.61±0.24 |
| TGN | 98.57±0.05 | 98.70±0.03 | 89.15±1.60 | 77.07±3.97 |
| TNCN | 98.60±0.02 | 98.89±0.03 | **92.77±0.07** | 92.81±0.08 |
| Inductive | | | | |
| Method | Wikipedia | Reddit | Mooc | Lastfm |
| CAWN | 98.24±0.03 | 98.19±0.03 | 81.42±0.24 | 89.42±0.07 |
| JODIE | 94.82±0.20 | 96.50±0.13 | 79.63±1.92 | 81.61±3.82 |
| DyRep | 92.43±0.37 | 96.09±0.11 | 81.07±0.44 | 83.02±1.48 |
| TGAT | 96.22±0.07 | 97.09±0.04 | 85.50±0.19 | 78.63±0.31 |
| NAT | 98.55±0.09 | 98.56±0.21 | 78.16±0.01 | 85.91±0.02 |
| TCL | 96.22±0.17 | 94.09±0.07 | 80.60±0.22 | 73.53±1.66 |
| DyGFormer | 98.59±0.03 | 98.84±0.02 | 86.96±0.43 | 94.23±0.09 |
| FreeDyG | **98.97±0.01** | **98.91±0.01** | 87.75±0.62 | 94.89±0.01 |
| EdgeBank | 80.63±0.00 | 85.48±0.00 | 49.43±0.00 | 75.49±0.00 |
| GraphMixer | 88.59±0.17 | 85.26±0.11 | 74.27±0.92 | 68.12±0.33 |
| TGN | 98.01±0.06 | 97.76±0.05 | 77.50±2.91 | 65.95±5.98 |
| TNCN | 98.31±0.05 | 98.43±0.39 | **91.56±0.23** | **95.74±0.50** |

Table 7: Comparison between TGN with heuristics and TNCN on tgbl-wiki Dataset

| Model | Val MRR | Test MRR | Training Time (s) | Inference Time (s) |
|---|---|---|---|---|
| TGN | 0.569 | 0.528 | 10.33 | 98.74 |
| TGN-CN | 0.561 | 0.505 | 12.33 | 106.21 |
| TGN-RA | 0.563 | 0.511 | 16.51 | 115.04 |
| TGN-AA | 0.565 | 0.517 | 11.42 | 115.01 |
| TGN-PPR | 0.521 | 0.427 | 207.01 | 327.22 |
| TGN-ELPH | 0.715 | 0.681 | 240.92 | 1614.86 |
| TNCN | 0.742 | 0.720 | 21.45 | 250.49 |

## C TNCN MODEL CONFIGURATION

**Network Choice** In our experiment, the changeable neural networks are chosen as follows:

In Memory Module, we choose $Identity$ as *msgfunc* and GRU as *upd*. In inference stage we process node memory with Graph Attention Embedding to get the temporal representation. As for Prediction Head, we finally choose $Linear$ as the $repr$ function.

**Hyper-parameter** Several detailed hyper-parameters for TNCN are shown in table 8, which can help researchers to reproduce the experiment performance as reported in this paper.

Table 8: Some Experiment Hyper-parameters

| Dataset | num_neighbors | num_epoch | patience | $mem\_dim$ | $emb\_dim$ | $time\_dim$ |
|---------|---------------|-----------|----------|-----------|-----------|------------|
| Wiki | 15 | 20 | 5 | 184 | 184 | 100 |
| Review | 15 | 10 | 3 | 184 | 184 | 100 |
| Coin | 10 | 5 | 3 | 100 | 100 | 100 |
| Comment | 10 | 3 | 2 | 100 | 100 | 100 |

**Parameter Analysis** We have also conducted experiments for parameter analysis. The results are shown in the table 9- 12.

Table 9: Performance metrics for different numbers of neighbors on Wiki dataset.

| num_neighbors | 10 | 12 | 15 | 18 | 20 |
|---------------|-----|-----|-----|-----|-----|
| Val MRR | 0.7433 | 0.7408 | 0.7412 | 0.7362 | 0.7418 |
| Test MRR | 0.7244 | 0.7193 | 0.7240 | 0.7187 | 0.7183 |
| training time (s) | 26.35 | 27.17 | 29.49 | 30.32 | 31.56 |
| test time (s) | 378.88 | 396.75 | 407.43 | 413.83 | 420.22 |

Table 10: Performance metrics for different numbers of neighbors on Coin dataset.

| num_neighbors | 5 | 8 | 10 | 12 | 15 |
|---------------|-----|-----|-----|-----|-----|
| Val MRR | 0.7492 | 0.7378 | 0.7430 | 0.7450 | 0.7406 |
| Test MRR | 0.7687 | 0.7619 | 0.7701 | 0.7662 | 0.7601 |
| training time (s) | 5936.19 | 6129.33 | 6406.00 | 6911.00 | 7529.01 |
| test time (s) | 56605.45 | 57117.99 | 57292.00 | 57745.00 | 57925.00 |

## D TIME CONSUMPTION STATISTICS

Table D exhibits the detailed time consumption on TGB datasets with different models. We can observe that TNCN maintains similar time consumption to memory-based networks while achieving striking speedup compared with graph-based models. All these experiments are conducted with NVIDIA GeForce RTX 3090.

Te be specific, we also conduct some experiments for the comparison between TNCN and NAT model. The hardware we use is NVIDIA GeForce RTX 2080 as NAT's code isn't compatible with 3090. 14 shows the final results. Note that NAT model exposes a backward as its instability, accomplishing about only 1/3 experiments when we test it.

Table 11: Performance metrics for different embedding and memory dimensions on Wiki dataset.

| emb_dim&mem_dim | 100 | 150 | 184 | 256 | 512 |
|---|---|---|---|---|---|
| Val MRR | 0.7402 | 0.7373 | 0.7412 | 0.7426 | 0.7404 |
| Test MRR | 0.7178 | 0.7159 | 0.7240 | 0.7224 | 0.7235 |
| training time (s) | 22.34 | 25.47 | 29.49 | 32.81 | 34.95 |
| test time (s) | 378.28 | 399.77 | 407.43 | 420.45 | 459.86 |

Table 12: Performance metrics for different embedding and memory dimensions on Coin dataset.

| emb_dim&mem_dim | 30 | 50 | 100 | 150 | 184 |
|---|---|---|---|---|---|
| Val MRR | 0.7387 | 0.7405 | 0.7430 | 0.7436 | 0.7518 |
| Test MRR | 0.7591 | 0.7606 | 0.7701 | 0.7646 | 0.7699 |
| training time (s) | 6228 | 6340 | 6406 | 6617 | 6721 |
| test time (s) | 56694 | 57179 | 57292 | 58103 | 59411 |

Table 13: Time Consumption of Different Models on TGB Datasets

| Model(tr/val/test)(s) | Wiki | Review | Coin | Comment | Flight |
|---|---|---|---|---|---|
| TGN | 23/117/124 | 2115/3731/3734 | 3259/6744/6352 | 8243/12580/12717 | 29681/52804/50147 |
| DyGFormer | 228/19801/19881 | 7760/57912/58011 | - | - | - |
| DyRep | 45/83/84 | 3257/3332/2990 | 4911/6382/6177 | 8921/11068/12701 | 31325/44917/47109 |
| TGAT | 160/9786/9861 | 4771/24854/25436 | - | - | - |
| CAWN | 301/24842/24851 | 11500/76094/76188 | - | - | - |
| TCL | 78/1623/1640 | 2344/3146/3148 | - | - | - |
| TNCN | 42/565/566 | 2076/9956/9957 | 5178/34294/31886 | 8732/38807/36642 | 34786/155231/144644 |

Table 14: Comparison of Time Consumption between TNCN and NAT

| Dataset | Model | Train (s) | Val (s) | Test (s) |
|---|---|---|---|---|
| tgbl-wiki | TNCN | 21.45 | 250.49 | 251.52 |
| | NAT | 74.92 | 298.6 | 298.41 |
| tgbl-review | TNCN | 1649 | 4788 | 4695 |
| | NAT | 422 | 7516 | 7461 |
| tgbl-coin | TNCN | 4920 | 28716 | 28805 |
| | NAT | 1896 | 30398 | 30176 |

---

**Algorithm 1** Pipeline of TNCN

---

1: **for** positive batch data $(posu, posv, t)$ **do**
2:     neg_batch $\leftarrow$ negative sampling
3:     **for** batch_data in {pos_data, neg_data} **do**
4:         mem, hist_events $\leftarrow$ memory_module(batch_data);     ▷ Get node memory and historical interactions
5:         emb $\leftarrow$ transform(mem, hist_events);                          ▷ Get the node embedding
6:         CN_mat $\leftarrow$ CN_extractor(hist_events);    ▷ Obtain the CNs for given node pair in a batch
7:         NCN_emb $\leftarrow$ AGG(emb, CN_mat);                      ▷ Aggregate the embeddings of CNs
8:         $p \leftarrow$ Pred(emb_u, emb_v, NCN_emb);          ▷ Calculate the probability of future links
9:     **end for**
10:     mem $\leftarrow$ memory_update(pos_data);                      ▷ Update the memory module
11: **end for**

---

## E  PSEUDOCODE OF TNCN PIPELINE

Algorithm 1 shows the pseudocode about the pipeline of our TNCN model.

## F  DETAILS OF COMMON NEIGHBOR EXTRACTION

Our temporal CN extractor begins with a sparse matrix $A$ constructed from the interactions of related nodes. We then include three stages to precisely generate arbitrary $(i, j)$-hop CNs:

1. Generate up to $k$-hop neighbors. The original matrix $A$ only includes 1-hop neighbors. To extend this, we: (a) Use self-loops for 0-hop neighbors, denoted as $A^0$. (b) Perform sparse matrix multiplication (e.g., $A^k$) to include arbitrary k-hop neighbors. Combining these two steps, we obtain an updated neighborhood matrix set $\hat{A} = \{A^i\}_{i=1}^K$.

2. Extract neighbors for each source and destination node with corresponding indices in the same batch. Assume that we require the $k$-th hop neighbors of node $u$, then vector $A^k[id(u)]$ is the result, where $id(u)$ stands for the reindexed id for node $u$. $A^k[id(u)][id(v)] = w > 0$ if $v$ is a $k$-hop neighbor of $u$, otherwise this element is $0$. $w$ represents the historical interaction frequency.

3. Obtain arbitrary $(i, j)$-hop CNs. We can perform hadamard product of $A^i[id(u)]$ and $A^j[id(v)]$ to acquire different hops of CNs. As the figure 4 shows, the operator can extract corresponding CNs for source-destination node pairs in a batch parallelly.

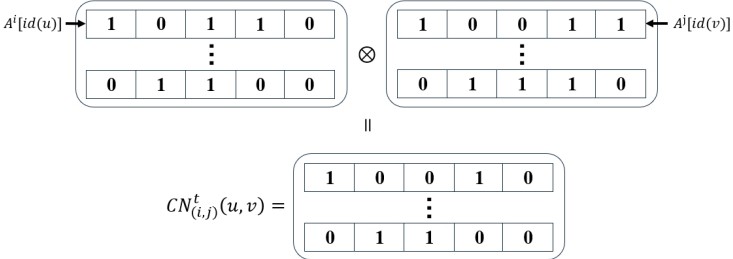

Figure 4: Common Neighbor Computation with Hadamard Product. Here node $\{1, 3, 4\}$ are $u$'s $i$-hop neighbors and $\{1, 4, 5\}$ are $v$'s $j$-hop neighbors. Thus the $(i, j)$-hop common neighbors between $u$ and $v$ are node $\{1, 4\}$.

By re-indexing the node IDs when generating $\hat{A}$ to prevent conflicts, the CN extractor can conduct the sparse matrix calculation, which are all performed in a Torch style that supports **batch operations**, thus enhancing parallelism and efficiency.

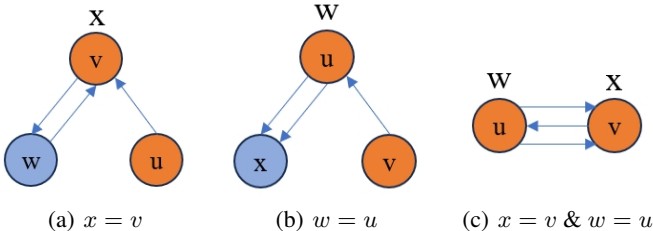

(a) $x = v$        (b) $w = u$        (c) $x = v$ & $w = u$

Figure 5: Here shows the special cases related to $(1, 2)$-hop CNs computation. Note that the graph is undirected, while the directed arrows implies the path direction used to determine the corresponding hop numbers.

### SPECIAL CASES ANALYSIS

Here are some special cases while calculating $(1, 2)$, $(2, 1)$ and $(2, 2)$ hop CNs. Under these situations, utilizing $A^k[id(u)]$ naively in step (2) will lead to walk-based neighbors, i.e. $\exists v, \exists i \neq j, w_i = w_j$, $s.t.\ (u, w_1, w_2, \cdots, w_{k-1}, v)\ exists$. To obtain a clear version of arbitrary path-based $(i, j)$-hop CNs, we should avoid the repetition of neighbors. We take $(1, 2)$ as an example to analyse the detailed method to eliminate repetition. Cases like $(2, 1)$ and $(2, 2)$ hop can be similarly solved.

Assume that node $x$ is a $(1, 2)$-hop CN of pair $(u, v)$, thus we know $\exists w,\ s.t.\ (u, x)$ and $(v, w, x)$ exist. There are two variants that render $x$ to be a walk-based CN instead of a path-based one that we exactly require.

(a) $x = v$. When $x = v$, the local graph has the topology shown in figure 5 a. This situation should satisfy two conditions: $w$ is a neighbor of $v$ and there are historical interactions between $u$ and $v$. Denote the frequency between $(u, v)$ before time $t$ as $q^t(u, v) = |\{(u, v, t')|t' < t\} \cup \{(v, u, t')|t' < t\}|$. So the naively computed $CN^t_{(1,2)}(u, v)[id(x)]$ value need to be subtracted by $[\sum_{w_i} q^t(w_i, v)] * q^t(u, v)$, i.e. the total interaction frequency of $v$ before time $t$ multiplied by the frequency between $(u, v)$.

(b) $w = u$. The structure is exhibited in figure 5 b. Here $(u, v)$ has historical edges and $x$ is a 1-hop neighbor of $u$. The additive substraction value is $[\sum_{x} q^t(x, u)^2] * q^t(u, v)$.

(c) Both (1a) and (1b) are satisfied. The ground truth is as figure 5 c. We just need to add back the overlap value that have been diminished once more.

Note that the procedure above can only deal with CNs of no more than $(2, 2)$-hop perfectly. For higher-order $(i, j)$-hop CN extraction, please refer to Perepechko & Voropaev (2009) for more details and complicated analysis.

## G   CASE STUDY

In this section figure 6, we show two case studies from real dataset of TGB, to give a better understanding of the effectivity of our TNCN.

Figure (a) shows a case from tgbl-wiki, which is a bipartite graph. The yellow nodes 0 and 15 are a $(u, v)$ pair. If we use a node-wise method to predict the future link of $(0, 15)$, we can find that node 15 has just 7 neighbors while node 0 has 12. So their properties may be different, thus having less chance to have an interaction. However TNCN can observe that the blue nodes are their $(1, 2)$-hop CNs and purple nodes are the $(2, 1)$-hop, and it will give a high probability over the existence of the future link.

Figure (b) shows another case from tgbl-coin. Here we need to predict the link of $(1, 8)$. Here TNCN can find that these two nodes have multiple variants of common neighbors. Node 3 is their $(1, 2)$-hop CN, node 4 and 6 are the $(2, 1)$-hop, and node 2 is both their $(1, 1)$ and $(2, 1)$ hop. The $(2, 2)$-hop

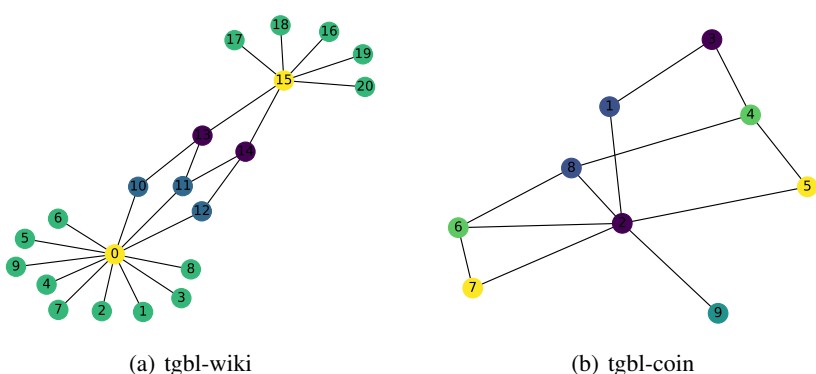

(a) tgbl-wiki            (b) tgbl-coin

Figure 6: Two case studies from TGB

CNs are node 5 and 7, while node 9 being a special $(2, 2)$-hop CN that owns a shared 1-hop node 2. With the abundant CN information, TNCN will be more likely to predict it as a positive future edge.

## H   PROOFS

In this section, we give proofs on theorems 5.4, 5.6 and 5.7.

We commence with a restatement of Theorem 5.4 for clarity:

**Theorem H.1.** *(Ability of encoding $k$-hop event). Given a $k$-hop event $\{(u_i, u_{i+1}, t_{u_i, u_{i+1}}) \mid i \in \{0, \ldots, k-1\}, k \geq 1\}$. If the node embedding of $u_0$ at time $t_{u_0, u_1}$, can be formally derived by the encoding function $Enc(\{(u_i, u_{i+1}, t_{u_i, u_{i+1}}) \mid i \in \{0, \ldots, k-1\}, k \geq 1\})$, then the learning method is considered capable of encoding the $k$-hop event. The following outline the encoding capabilities of different learning schemes:*

- *Memory-based approach can encode any $k$-hop events with $k = 1$.*

- *Memory-based approach can encode any monotone $k$-hop events with arbitrary $k$.*

- *$k$-hop-subgraph-based approach can encode any $k'$-hop events with $k' \leq k$*

*Proof.* In the following analysis, we establish the encoding efficacy of the memory-based approach. Consider a $k$-hop event with the simplifying assumption that $k = 1$, which reduces the event to the tuple $(u_0, u_1, t_{u_0, u_1})$. By adhering to the predefined schematics of the memory-based methodology, the memory state $Mem(u_0, t_{u_0, u_1})$ is updated via the function $f_{mem}$ such that $Mem(u_0, t_{u_0, u_1}) = f_{mem}(Mem(u_0, t'), e_{u_0, u_1}^{t_{u_0, u_1}}, t_{u_0, u_1} - t')$. Let us denote the encoding function as $Enc = f_{mem}(Mem(u_0, t'), \ldots)$. It is our intention to demonstrate that this memory-based framework is capable of encoding any $k$-hop event for $k = 1$.

We consider the encoding of an arbitrary monotonically increasing $k$-hop temporal event sequence within a memory-based approach. The induction principle is applied to demonstrate the capability of this approach. For the base case, $k = 1$, the encoding has been shown to be feasible. Now, assume the proposition holds for a $k'$-hop event; that is, any $k'$-hop temporal sequence of monotonically increasing events can be encoded using a memory-based approach. This assumption implies that there exists an embedding function such that

$$Emb(u_0, t_{u_0, u_1}) = Enc((u_i, u_{i+1}, t_{u_i, u_{i+1}}) \mid i \in 0, \ldots, k'-1), \tag{8}$$

for all event sequences with $k'$ hops, where $k' \geq 1$. Given an arbitrary $k' + 1$-hop event, which can be partitioned into an initial event $(u_0, u_1, t_{u_0, u_1})$ and a subsequent $k'$-hop sequence. The existence

of an encoding function for the $k'$-hop sequence assures that

$$
\begin{aligned}
Emb(u_0, t_{u_0,u_1}) &= f_{emb}(Mem(u_0, t_{u_1,u_2})) \\
&= f_{emb}(f_{mem}(Mem(u_0, t_{u_1,u_2}), Mem(u_1, t_{u_1,u_2}), e_{u_0,u_1}^{t_{u_0,u_1}}, t_{u_0,u_1} - t_{u_1,u_2}), \\
Mem(u_1, t_{u_1,u_2}) &= f_{emb}^{-1}(Emb(u_1, t_{u_1,u_2})) \\
&= f_{emb}^{-1}(Enc(\{(u_i, u_{i+1}, t_{u_i,u_{i+1}}) \mid i \in \{1, \ldots, k'\}, k' \geq 1\}))
\end{aligned}
\tag{9}
$$

Subsequently, it is demonstrated that $Emb(u_0, t_{u_0,u_1})$ provides an encoding for both the initial event and the $k'$-hop sequence, thereby affirming its efficacy in encoding the entire $k' + 1$-hop event. This concludes the inductive step and substantiates the inductive argument.

We consider a $k$-hop-subgraph-based approach for our analysis. It is evident that a $k$-hop subgraph encompasses any $k'$-hop events, where $k' \leq k$. Furthermore, the aggregation methodology assimilates all nodes contained within the subgraph. Collectively, these observations substantiate the theorem in question.

$\square$

We commence with a restatement of Theorem 5.6 for clarity:

**Theorem H.2.** *(Learning method time complexity). Denote the time complexity of a learning method as a function of the total number of events processed during training. For a given graph $\mathcal{G}$ with the number of nodes designated as $|\mathcal{N}|$ and the number of edges as $|\mathcal{E}|$, the following assertions hold:*

- *For the memory-based approach, the time complexity is $\Theta(|\mathcal{E}|)$.*

- *For $k$-hop-subgraph-based with $k = 1$, the lower-bound time complexity is $\Omega\left(\frac{|\mathcal{E}|^2}{|\mathcal{N}|}\right)$, and the upper-bound time complexity is $\mathcal{O}\left(\frac{|\mathcal{E}|^2}{|\mathcal{N}|} + |\mathcal{E}||\mathcal{N}|\right)$*

- *For $k$-hop-subgraph-based with $k = 2$, the upper-bound time complexity is $\mathcal{O}\left((\frac{|\mathcal{E}|^2}{|\mathcal{N}|} + |\mathcal{E}||\mathcal{N}|)^{\frac{3}{2}}\right)$*

*Proof.* In the proposed theorem, the time complexity is denoted as the aggregate quantity of events processed throughout the training phase. The objective herein is to ascertain the precise count of such utilized events.

In the context of the memory-based methodology, it is evident that each event is utilized a singular time. Consequently, the cumulative number of events is expressed as $|\mathcal{E}|$, which infers that the time complexity adheres to the order of $\Theta(|\mathcal{E}|)$.

In the context of $k$-hop-subgraph-based algorithms wherein $k = 1$, an event $(u, v, t)$ is exploited once for every incident event within the neighborhood of vertices $u$ or $v$. Without loss of generality, we focus on all events within the 1-hop-subgraph of vertex $u$. The aggregate count of events processed is given by $\sum_{i=1}^{d(u)} i = \Theta\left(d(u)^2\right)$, where $d(u)$ denotes the degree of vertex $u$. Consequently, the computational complexity is fundamentally proportional to $\sum_{u \in \mathcal{N}} d(u)^2$. Drawing on the results of de Caen (1998), the lower bound on the time complexity is established as $\Omega\left(\frac{|\mathcal{E}|^2}{|\mathcal{N}|}\right)$, whereas the upper bound is determined as $\mathcal{O}\left(\frac{|\mathcal{E}|^2}{|\mathcal{N}|} + |\mathcal{E}||\mathcal{N}|\right)$

In the context of $k$-hop-subgraph-based algorithms wherein $k = 2$,, we adopt similar strategy where each event $(u, v, t)$ will only be utilized once another event within the subgraph of $u$ or $v$ is firstly considered. The total number of events can be formulated as $\sum_{u \in \mathcal{N}} d(u) \sum_{v \in \mathcal{N}_u} \sum_{w \in \mathcal{N}_v} d(w)$. Replacing $d(u)$ as $X_i$, $\sum_{v \in \mathcal{N}_u} \sum_{w \in \mathcal{N}_v}$ as $Y_i$, we reformulated is as $\sum_{i \in |\mathcal{N}|} X_i Y_i$, satisfying $\sum_{i \in |\mathcal{N}|} X_i^2 = \sum_{u \in \mathcal{N}} d(u)^2$ and $\sum_{i \in |\mathcal{N}|} Y_i^2 = \sum_{u \in \mathcal{N}} d(u)^4$. Following Cauchy inequality and conclusions of $\sum_{u \in \mathcal{N}} d(u)^2$, we got the the upper-bound time complexity is $\mathcal{O}\left((\frac{|\mathcal{E}|^2}{|\mathcal{N}|} + |\mathcal{E}||\mathcal{N}|)^{\frac{3}{2}}\right)$ $\square$

We commence with a restatement of Theorem 5.7 for clarity:

**Theorem H.3.** *TNCN is strictly more expressive than CN, RA, and AA.*

We first give definitions of these structural features under temporal settings. Given two nodes $u$ and $v$, the structural features before time $t$ are defined as follows:

$$CN(u,v,t) = \sum_{w \in N_1^t(u) \cap N_1^t(v)} 1,$$

$$RA(u,v,t) = \sum_{w \in N_1^t(u) \cap N_1^t(v)} \frac{1}{d(w)}, \quad (10)$$

$$AA(u,v,t) = \sum_{w \in N_1^t(u) \cap N_1^t(v)} \frac{1}{\log d(w)}$$

Given a node $u$, the degree of node $u$ is the number of events $e$ with an endpoint at node $u$. Without loss of generality (WLOG), we consider node $u$ as the source node, and the events are $\{(u, v_i, t_i) \mid i \in \{0, \ldots, k-1\}, k \geq 1\}$. Each time a new event is given, the embedding of node $u$ is updated by

$$mem_u^{t_i} = upd_{src}(mem_u^{t_{i-1}}, msgfunc_{src}(e_{u,v_i}^{t_i})). \quad (11)$$

With the MPNN universal approximation theorem, *msgfunc* can be a constant function, and $upd_{src}$ can be an addition function. Thus,

$$mem_u^{t_{k-1}} = d(u). \quad (12)$$

Then the embedding can learn arbitrary functions of node degrees, i.e.,

$$emb_u^t = f(d(u)). \quad (13)$$

Thus, the neural common neighbor $TNCN_1(u,v) = \oplus_{w \in N_1^t(u) \cap N_1^t(v)} emb_w^t$ can express Equation 10.

Extending to situations where the common neighbor node has some features we want to learn, the traditional CN, RA, and AA cannot accommodate this. However, our TNCN can express these features, demonstrating that TNCN is strictly more expressive than CN, RA, and AA.

