# OpenReview forum: "Efficient Neural Common Neighbor for Temporal Graph Link Prediction"
_ICLR.cc/2025/Conference — Submitted to ICLR 2025_

### Official Review · Reviewer_Erh5 · 2024-10-21

**Soundness:** 3
**Presentation:** 3
**Contribution:** 3
**Rating:** 6
**Confidence:** 4

**Summary:**

The paper is about temporal link prediction, which is an interesting topic. The authors propose a temporal version of NCN for link prediction in temporal graphs, which dynamically updates a temporal neighbor dictionary for each node and utilizes multi-hop common neighbors between the source and target node to learn a more effective pairwise representation. The paper is well written and well organized. However, there are several concerns in the current version of the paper that addressing them will increase the quality of this paper.

**Strengths:**

1 Cutting-edge research directions.

2 Clear writing logic.

3 Sufficient experimental results.

**Weaknesses:**

1 The authors should have a special discussion on whether the biggest difference between TNCN and NCN is the core contribution of this paper. If so, this should be highlighted. If not, more introduction is needed on the importance of the new scenario.

2 Since the strategy proposed in the paper is built around the batch processing mode of sequential graph learning, whether the batch size will have a different impact on the strategy is something that needs to be considered and discussed.

3 The motivation and contribution of the paper are worthy of recognition, but in the main text, the authors can consider putting more emphasis on the contribution description and logical arrangement. At present, it seems that the proof takes up a certain amount of space, making the method and experiment part seem less substantial, and the information that can be expressed is not clear and comprehensive enough.

4 The authors could consider discussing the computational complexity, especially comparing it to similar methods (including static graphs and temporal graphs).

**Questions:**

As above.

---

> ### Author Response · Authors · 2024-11-21
>
> Thank you for acknowledging the effectiveness and efficiency of our method demonstrated by extensive experiments. We address your concerns as follows.
>
> #### Weaknesses:
> > W1. The authors should have a special discussion on whether the biggest difference between TNCN and NCN is the core contribution of this paper. If so, this should be highlighted. If not, more introduction is needed on the importance of the new scenario.
>
> Thank you for the suggestion on clarifying our contributions. In fact, the core contribution is our combination of memory-based backbone and Neural Common Neighbor method for temporal link prediction, which is exactly about the difference between our TNCN and traditional NCN.
>
> For the utilization of memory-based backbone, it uses a memory module to dynamically store and update the node memory. This pipeline can process the edge interaction stream only once for inference, avoiding the repeated computation of node embeddings within different subgraph. However, traditional NCN uses static GNNs as its backbone, thus bringing about high computational costs. Another is our NCN extractor. Here we adopt a batch-wise operation to extract the common neighbor information between a given node pair. These information can be used to aggregate the related node embeddings to obtain the pair-wise features, thus enhancing the capability of our TNCN model.
>
> For a clear representation, we have also appended a table about the comparison between TNCN and NCN in section 4 in the revised PDF. We have also put it below:
> | Model | Temporal Scenario | Backbone        | Arbitrary CN Hops | Batch-wise CN Extraction |
> |-------|-------------------|-----------------|-------------------|--------------------------|
> | NCN   | ❌                | Traditional GNN | ❌                | ❌                       |
> | TNCN  | ✔️                | Memory-based    | ✔️                | ✔️                       |
>
> > W2. Since the strategy proposed in the paper is built around the batch processing mode of sequential graph learning, whether the batch size will have a different impact on the strategy is something that needs to be considered and discussed.
>
> Thank you for your inquiry regarding batch size. In our experiments, two notations of batch size should be considered: the ''benchmark batch size'' and the ''model batch size''.
> 	1. The benchmark batch size refers to how the benchmark processes events. Theoretically, each event can access all preceding events for prediction. However, to reduce computational cost and seek for a fair evaluation comparison, most benchmarks process consecutive events that occur in a specified time interval, allowing the model to make predictions for them using the same context. In TGB, this consecutive grouping is fixed at 200 for all models.
> 	2. The model batch size defines the number of events the model exactly processes in a batch. In TGB, this is also commonly set to 200, which we follow in our default configuration.
>
> We also conducted an ablation study to analyze the effects of varying the model batch size. Our results indicate that our model achieves consistent, linear speedup with increasing batch size, showing potential for further scaling. In contrast, some models that lack batch processing capabilities do not demonstrate this acceleration.
>
> | Wiki\batchsize | 50 | 100 | 200 | 500 | 1000 |
> |---|---|---|---|---|---|
> | training time(s) | 88.93  | 44.69  | 29.49  | 14.07  | 9.14  |
> | test time(s) | 393.70  | 419.46  | 407.43  | 417.50  | 385.09  |
> | val mrr | 0.7705  | 0.7562  | 0.7412  | 0.7177  | 0.6940  |
> | test mrr | 0.7484  | 0.7345  | 0.7240  | 0.7057  | 0.6879  |
>
> | Review\batchsize | 50 | 100 | 200 | 500 | 1000 |
> |---|---|---|---|---|---|
> | training time(s) | 1481  | 1114  | 780  | 576  | 422  |
> | test time(s) | 2430  | 2434  | 2562  | 2703  | 2899  |
> | val mrr | 0.3607  | 0.3602  | 0.3584  | 0.3546  | 0.3519  |
> | test mrr | 0.4302  | 0.4231  | 0.4190  | 0.3966  | 0.3967  |
>
> For W3 and W4, please see the comment below.

---

> > ### Author Response · Authors · 2024-11-21
> >
> > > W3. The motivation and contribution of the paper are worthy of recognition, but in the main text, the authors can consider putting more emphasis on the contribution description and logical arrangement. At present, it seems that the proof takes up a certain amount of space, making the method and experiment part seem less substantial, and the information that can be expressed is not clear and comprehensive enough.
> >
> > Thank you for your question regarding to our paper organization about the motivation and contributions. Note that there is no proof in the main text, but definition and problem formulation. Our primary goal is to provide an efficient and effective solution, validated in large-scale settings in terms of both performance and resource consumption. Given that our model is an instance of a broader framework, we also want to highlight the consideration including different frameworks that could support future research in this area, thus making some formulations about dynamic graph modeling.
> >
> > We appreciate your feedback and will consider condensing these sections while moving some of the experiments from the appendix into the main content to enhance clarity.
> >
> > > W4. The authors could consider discussing the computational complexity, especially comparing it to similar methods (including static graphs and temporal graphs).
> >
> > Thank you for your suggestion regarding computational complexity. By analyzing the time complexity of prior works and formalizing a unified comparison framework, we obtained the following results:
> >
> > Here we use some notations.
> > $k$: hop, $N$: number of neighbors, $L$: sampling walk length, $c$: scaling value.
> >
> > |Methods|Feature computation|Auxiliary computation|
> > |---|---|---|
> > |TGN|$cN$|/|
> > |TGAT|$cN^k$|/|
> > |CAWN|$cL^k$|$L^k$|
> > |DyGFormer|$c(N + N^2)$|$N$|
> > |TNCN|$cN$|$N^k$|
> >
> > Feature computation counts the number of embeddings fed into the model, reflecting the complexity of the sampled graph to the static/dynamic GNN. Auxiliary computation counts additional operations needed such as proposing candidate annoymous walk or retrieving common neighbor embeddings. Note that the total complexity cannot be computed by simply adding them. Instead, feature computation has a **large constant** value $c$ that relates to model training/inference, embedding/memory dimension, etc., while auxiliary computation is usually very cheap.
> >
> > Most theoretical results align well with the practical computation costs illustrated in Figure 3 of our paper. However, certain works, such as DyGFormer, exhibits lower theoretical complexity than TGAT but incurs higher practical costs. Our investigation revealed that this discrepancy is due to the slower implementation of encoding neighbor co-occurrence.
> >
> > We have also shown the exact time consumption of these methods on different datasets. Please refer to Appendix D for details.

---

> > > ### Comment · Reviewer_Erh5 · 2024-11-23
> > >
> > > Thank you for the authors' reply. I think this paper has some contribution and can be considered for acceptance. I will maintain my positive score.

---

> > > > ### Author Response · Authors · 2024-11-23
> > > >
> > > > Thank you for your positive feedback! We are delighted to your acknowledgement about our contribution. If you have any concerns or wish to engage in future discussion, please do not hesitate to reach out.

---

### Official Review · Reviewer_P1SL · 2024-10-27

**Soundness:** 3
**Presentation:** 3
**Contribution:** 2
**Rating:** 5
**Confidence:** 4

**Summary:**

The author focuses on link prediction tasks in temporal graphs, which have ubiquitous applications in real-world systems. Instead of implicitly encoding common neighbor features from the historical neighborhood of the target node, the author proposes the Temporal Common Neighbor Extractor to explicitly integrate these features into the process of temporal graph representation learning, achieving both effectiveness and efficiency compared to existing methods on the TGB benchmark. Additionally, the author provides theoretical analysis based on their proposed method, offering a more thorough illustration of the model.

**Strengths:**

This paper has several strengths worth noting:

* **Interesting Motivation.** The motivation for extracting common neighbor features is compelling.
* **Extensive Experiments.** The author has designed a variety of experiments to demonstrate the effectiveness and efficiency of their methods.
* **Well-Organized Representation.** The paper is well-structured, and the theoretical analysis provides strong support for the proposed methods.

**Weaknesses:**

However, the paper also has some weaknesses, outlined as follows:

* **Lack of Novelty.** Firstly, the idea of extracting common neighbors in temporal graphs has certainly been explored before. It seems that the design of your key component, the "CN Extractor," closely resembles existing work in KDD2024 [1]. Moreover, your CN extracting component does not appear to include any specific improvements for temporal graphs. Simply extracting "monotone k-hop events" does not substantiate this claim.
* **Lack of Important Baselines.** Since you "extend Neural Common Neighbor for static prediction methods," these static methods should also be included in your comparisons.
* **Lack of Case Study.** Providing specific case studies could enhance the understanding of your method.
* **Ambiguous Expression.** What does $emb$ mean in Equation 2? It seems you haven't explained it anywhere—does it refer to memory? Your method encodes the common neighbor neighborhood composed of source-destination node pairs; might this lead to a loss of other information (e.g., nodes that are not common)?
* **Parameter Analysis.** Given that your method is based on multi-hop common neighbors, analyzing parameters across different multi-hop settings could better validate the robustness of your model.

[1] Co-Neighbor Encoding Schema: A Light-cost Structure Encoding Method for Dynamic Link Prediction, KDD 2024.

**Questions:**

Q1: In what ways does your approach build upon or differ from existing methods that also extract common neighbors in temporal graphs?

Q2: Can you clarify the importance of including static baselines in your experiments, and how their absence affects the interpretation of your results?

Q3: What specific case studies can you provide to illustrate the practical application and effectiveness of your proposed method?

Q4: Can you provide a detailed explanation of the notation used in your equations, particularly regarding $emb$ in Equation 2, and how this impacts the overall methodology?

Q5: How do you plan to conduct a thorough parameter analysis for the multi-hop common neighbors, and what insights do you anticipate this will provide regarding your model's robustness?

---

> ### Author Response · Authors · 2024-11-21
>
> Thank you for acknowledging the motivation and representation, and the effectiveness and efficiency of our method demonstrated by extensive experiments. We address your concerns as follows.
>
> #### Weaknesses:
> > W1. Lack of Novelty. Firstly, the idea of extracting common neighbors in temporal graphs has certainly been explored before. It seems that the design of your key component, the "CN Extractor," closely resembles existing work in KDD2024 [1]. Moreover, your CN extracting component does not appear to include any specific improvements for temporal graphs. Simply extracting "monotone k-hop events" does not substantiate this claim.
>
> Thanks for recommending this contemporaneous work. This paper is on arxiv in July and published in KDD in August, while ICLR submssion is on October 1st. It is within four months and is not required to be compared according to ICLR policy (https://iclr.cc/Conferences/2025/FAQ). Nevertheless, we have incorporated the comparison in the revision.
>
> For the aspect of encoding common neighbor information, TNCN has its own distinctions apart from CNE-N from [1]. The biggest difference between our TNCN and the CNE-N is that, we **explicitly utilize the node embeddings** of the common neighbors in the prediction stage. While CNE-N calculates the co-neighbor encoding of node pairs within the local subgraph, it only records the number of the neighboring nodes, ignoring their embedding information. There are also two slight differences between the two models. Given node pair $(u,v)$, CNE-N considers the number of common neighbors of $(u,i)$ and $(v,i)$ pairs for all $i$ in the local subgraph. In contrast, TNCN only takes the common neighbors between $u$ and $v$ and their embeddings into account. Furthermore, CNE-N uses hash table to map a node to its position while TNCN takes monotone storage to keep the order of the input events. Our implementation is more friendly when it's required to take the most recent interactions within a specified time interval.
>
> > W2. Lack of Important Baselines. Since you "extend Neural Common Neighbor for static prediction methods," these static methods should also be included in your comparisons.
> Lack of Case Study. Providing specific case studies could enhance the understanding of your method.
>
> #### About static methods
>
> Thanks for your suggestion on adding the static methods as baselines.
>
> In fact, we had put the experimental results of some classic static methods in appendix B.2. We considered CN, RA, AA, PPR and ELPH of BUDDY. Please refer to the table 7 for detailed performances.
>
> #### About case studies
> Thanks for your advice. You can refer to the figure 1 in our paper to get a first sight about the effectivity of our TNCN. We have also complemented some case study examples from Wiki and Coin dataset to make it more intuitive to understand the key component. Please refer to Appendix G in our revised PDF.
>
> > W3. Ambiguous Expression. What does 'emb' mean in Equation 2? It seems you haven't explained it anywhere—does it refer to memory? Your method encodes the common neighbor neighborhood composed of source-destination node pairs; might this lead to a loss of other information (e.g., nodes that are not common)?
>
> Thanks for your point. Here the 'emb' refers to the node embeddings, which can be obtained from the node memory (i.e. 'mem') with some transformations, including identity, linear, or graph attention, etc. Our implementation follows the standard TGN, which uses graph attention. Thus, emb also includes neighbors' mem information. We have updated it in the revised PDF.
>
> Note that TNCN does not only leverage common neighbor while losing other information (e.g., nodes that are not common). As shown in Equation (5) in the paper, the final representation of (u,v) combines both u and v's embeddings as well as their common neighbors' embeddings, where u and v's embeddings contain those non-common-neigbhor information.
>
> For W4, W5 and Questions, please see the comment below.

---

> > ### Author Response · Authors · 2024-11-21
> >
> > > W4. Parameter Analysis. Given that your method is based on multi-hop common neighbors, analyzing parameters across different multi-hop settings could better validate the robustness of your model.
> >
> > As per you request, we conduct experiments for the analysis of the hyperparameters to better validate the robustness and capability of our model. We particularly focus on NCN-hop-number, num_neighbors and emb_dim, which have a larger influence on the performance of the model. The experimental results are shown as follows:
> > | Wiki/num_neighbors | 10 | 12 | 15 | 18 | 20 |
> > |---|---|---|---|---|---|
> > | val mrr | 0.7433  | 0.7408  | 0.7412  | 0.7362  | 0.7418  |
> > | test mrr | 0.7244  | 0.7193  | 0.7240  | 0.7187  | 0.7183  |
> > | training time(s) | 26.35  | 27.17  | 29.49  | 30.32  | 31.56  |
> > | test time(s) | 378.88  | 396.75  | 407.43  | 413.83  | 420.22  |
> >
> > | Coin/num_neighbors | 5 | 8 | 10 | 12 | 15 |
> > |---|---|---|---|---|---|
> > | val mrr | 0.7492  | 0.7378  | 0.7430  | 0.7450  | 0.7406  |
> > | test mrr | 0.7687  | 0.7619  | 0.7701  | 0.7662  | 0.7601  |
> > | training time(s) | 5936.19  | 6129.33  | 6406.00  | 6911.00  | 7529.01  |
> > | test time(s) | 56605.45  | 57117.99  | 57292.00  | 57745.00  | 57925.00  |
> >
> > | Wiki/emb_dim&mem_dim  | 100 | 150 | 184 | 256 | 512 |
> > |---|---|---|---|---|---|
> > | val mrr | 0.7402  | 0.7373  | 0.7412  | 0.7426  | 0.7404  |
> > | test mrr | 0.7178  | 0.7159  | 0.7240  | 0.7224  | 0.7235  |
> > | tr time | 22.34  | 25.47  | 29.49  | 32.81  | 34.95  |
> > | test time | 378.28  | 399.77  | 407.43  | 420.45  | 459.86  |
> >
> > | Coin/emb_dim&mem_dim | 30 | 50 | 100 | 150 | 184 |
> > |---|---|---|---|---|---|
> > | val mrr | 0.7387  | 0.7405  | 0.7430  | 0.7436  | 0.7518  |
> > | test mrr | 0.7591  | 0.7606  | 0.7701  | 0.7646  | 0.7699  |
> > | tr time | 6228  | 6340  | 6406  | 6617  | 6721  |
> > | test time | 56694  | 57179  | 57292  | 58103  | 59411  |
> >
> > We have also attached these parameter analysis in the appendix C in our revised PDF, and we will make a better organization later.
> >
> > [1] Co-Neighbor Encoding Schema: A Light-cost Structure Encoding Method for Dynamic Link Prediction, KDD 2024.
> >
> > #### Questions:
> > > Q1. In what ways does your approach build upon or differ from existing methods that also extract common neighbors in temporal graphs?
> >
> > Please refer to the response to W1.
> >
> > > Q2. Can you clarify the importance of including static baselines in your experiments, and how their absence affects the interpretation of your results?
> >
> > Please refer to the response to W2 part 1.
> >
> > > Q3. What specific case studies can you provide to illustrate the practical application and effectiveness of your proposed method?
> >
> > Please refer to the response to W2 part 2.
> >
> > > Q4. Can you provide a detailed explanation of the notation used in your equations, particularly regarding 'emb' in Equation 2, and how this impacts the overall methodology?
> >
> > Please refer to the response to W3.
> >
> > > Q5. How do you plan to conduct a thorough parameter analysis for the multi-hop common neighbors, and what insights do you anticipate this will provide regarding your model's robustness?
> >
> > Please refer to the response to W4.

---

> > > ### Author Response · Authors · 2024-11-25
> > > **Looking forward to your response!**
> > >
> > > Dear Reviewer,
> > >
> > > The discussion period is nearly ending soon, and we wonder whether our rebuttal is sufficient to address your concerns.
> > >
> > > If you have any further questions or wish to engage in further discussion, please do not hesitate to reach out. And if you find the content in our main paper and rebuttal satisfactory, would you please consider raising your score?

---

> > > > ### Author Response · Authors · 2024-11-28
> > > > **Looking forward to your response!**
> > > >
> > > > Dear reviewer,
> > > > Thanks for your comments. In our rebuttal, we have clarify the distinction between CNE-N and our TNCN model, and add relative comparison in the revised PDF, even if this contemporaneous work is not required to be compared according to ICLR policy. The comparison with static methods can be referred to appendix in the original paper while the case studies, the parameter analysis are both complemented in the revised PDF now. And we have also clarified some notations in the revision.
> > > >
> > > > The discussion period is going to end soon, and we wonder whether our rebuttal is sufficient to address your concerns. If you find the content in our main paper and rebuttal satisfactory, would you please consider raising your score?

---

### Official Review · Reviewer_BFZ2 · 2024-10-31

**Soundness:** 2
**Presentation:** 1
**Contribution:** 1
**Rating:** 3
**Confidence:** 4

**Summary:**

This paper proposes a method for link prediction on continuous-time dynamic graphs (CTDGs), aiming to unify two prominent model families: memory-based models and neighbor-based models. The authors introduce TNCN, demonstrating experimentally that it performs better in certain configurations and is more efficient than existing models in the literature.

**Strengths:**

- **Evaluation**: The proposed model is evaluated on established benchmarks, enhancing the reliability of the results.

- **Engineering**: The paper introduces an engineering approach to combine common neighbor (CN) techniques with memory-based methods, integrating these two modeling approaches.

**Weaknesses:**

**Presentation**: The abstract implies that common neighbor methods are primarily used in static graphs, overlooking their established role in dynamic graph modeling. Additionally, the motivation for combining memory-based and neighbor-based techniques is presented only briefly at the end of the introduction.

**Limited Novelty**: The proposed model's core components primarily consist of established techniques. For instance, the memory-based module closely resembles TGN, lacking additional innovations or a clear positioning relative to other memory-based approaches. Similarly, while TNCN incorporates common neighbor (CN) techniques with optimized computation, it employs multi-hop methods similar to those in models like CAWN, further limiting its novelty.

**Theoretical Claims**: The paper presents familiar results as novel contributions, such as the $O(∣E∣)$ memory complexity for event aggregation, along with upper and lower bounds that are established in prior work (e.g., Caen, 1998). Presenting these as new theoretical results may be misleading.

The paper exhibits strong engineering but lacks clear new contributions. Theoretical results on memory complexity and specific experimental results are weak, with minimal performance gains in dynamic link prediction tasks. Given the limitations in presentation, contribution, and experimental validation, I recommend rejection.

**Experimental results**: While combining memory and neighbor-based methods is interesting, the observed performance gains are minimal, with certain results (e.g., in Table 5) showing limited improvement using unclear metrics (possibly AUC or AP).

**Questions:**

1. Could you clarify the main novelty of your memory-based module and its differentiation from TGN?
2. What is the specific metric used in Table 5? Understanding whether it's AUC or AP is essential for interpreting the results.

---

> ### Author Response · Authors · 2024-11-21
>
> Thank you for acknowledging our reliability of evaluation and the combination of the modeling approches. We address your concerns as follows.
>
> #### Weaknesses
> > W1. Presentation: The abstract implies that common neighbor methods are primarily used in static graphs, overlooking their established role in dynamic graph modeling. Additionally, the motivation for combining memory-based and neighbor-based techniques is presented only briefly at the end of the introduction.
>
> Thank you for your comment on our presentation. Firstly, we want to clarify that we do not intend to overlook previous work; in fact, we provide a comprehensive discussion on related work in both Sections 1 and 2. Below we further explain each point in detail.
>
> Abstract
>
> Note that we said "traditional methods usually ..." in abstract. That is exactly the case. The methods in large-scale benchmark TGB, including DyRep, TGN, CAWN, TGAT, TCL, Edgebank and Graphmixer, primarily still do not involve pair-wise information to make the final prediction. Only two out of nine methods consider pairwise features in dynamic graph modeling, thus substantiating our statement.
>
> A second perspective, which we did not emphasize in the abstract, is how our model achieves greater efficiency than graph-based framework methods that already incorporate pairwise information. Balancing the use of pairwise information and the overall efficiency is essential but challenging. Pairwise information was typically excluded in memory-based methods due to its high computational cost, and existing dynamic graph models face limitations in handling it. For instance, DyGFormer only considers the frequency of 1-hop common neighbors without capturing their features or embeddings, while NAT learns conditional representations for each node’s neighbors, requiring subsampling to avoid the quadratic complexity. These limitations make these models less applicable to large datasets. To address these challenges, we developed TNCN, which is an efficient and powerful pairwise model.
>
> We chose not to highlight all the points in the abstract to present our model more straightforwardly, not to overlook existing methods. We certainly acknowledge the contributions of prior work, as outlined in our Introduction (lines 80-85) and in Section 2.2 (Related Work). With your feedback in mind, we have further revised the abstract to clarify these dual perspectives. We welcome any suggestions on how we might better convey our approach while balancing simplicity and comprehensiveness.
>
> Motivation
>
> Our motivation of combining memory-based backbone and neighbor-based technique is not limited to the end of the introduction. In fact, in section 1 (introduction) paragraph 3 and 4, we have first mentioned that we are urged to seek for pair-wise information for the innate drawbacks of node-wise prediction. In paragraph 5, we have stated that graph-based methods commonly suffer from high computational complexity issue, thus turning our sight to the more efficient memory-based backbone. In section 2 (related work) we have also talked about the characteristics about memory-based methods with efficiency and graph-based with pair-wise feature, enhancing our motivation about our innovation on combining the memory-based backbone and dynamic NCN processor. We will present the introduction better to make the motivation more clear.
>
> For other concerns, please see the following comments.

---

> > ### Author Response · Authors · 2024-11-21
> >
> > > W2. Limited Novelty: The proposed model's core components primarily consist of established techniques. For instance, the memory-based module closely resembles TGN, lacking additional innovations or a clear positioning relative to other memory-based approaches. Similarly, while TNCN incorporates common neighbor (CN) techniques with optimized computation, it employs multi-hop methods similar to those in models like CAWN, further limiting its novelty.
> >
> > Thank you for your question on how our model differs from previous approaches. The major novelty of TNCN lies in **combining the memory-based pipeline with the NCN method** to leverage pairwise information in an efficient way. Despite its simplicity, **no previous work has attempted in this direction**; at the same time, we achieved **new state-of-the-art results** on 3 out of 5 large-scale TGB datasets.
> >
> > Below, we will further address the novelty concern from two perspectives:
> >
> > Memory-Based Backbone
> >
> > Note that our TNCN model adopts the memory-based backbone following the traditional memory-based pipeline. This pipeline utilizes a memory module to store and update the node memory, which can be transformed into the embeddings for the final prediction or downsteaming tasks. There are two representative classes of memory-based methods, DyRep and TGN, which actually use the same pipeline. They only have minor differences: DyRep uses old memory to obtain the embeddings, while TGN first updates the memory with new information. Other differences are about the concrete functions for the memory update, where DyRep and TGN utilize RNN and GRU respectively. Since advancing memory-based pipeline is not the focus of our paper, we directly adopt the TGN backbone. The significant improvement of TNCN over TGN highlights the effectiveness of combining NCN with memory-based pipeline. Note that TGN was proposed 4 years ago, thus it is highly possible that the TNCN's performance can be further improved with advanced memory-based backbones.
> >
> > Multi-Hop Methods
> >
> > TNCN’s way of using multi-hop neighbors distinguishes it from other models like CAWN and NAT:
> > 	1.	CAWN uses **causal anonymous walks** to capture node occurrences within walks, constructing representations for node pairs $(u, v)$. However, this method involves **node sampling** to select specific anonymous walks, increasing computational complexity due to repeated resampling and introducing randomness and uncertainty. Additionally, CAWN enforces **monotonically decreasing timestamps** on walks, which further limits the model’s representational capacity.
> > 	2.	NAT builds a dictionary for each node to store neighbor representations conditioned on that node. For prediction, NAT reconstructs the joint neighborhood from the two nodes’ dictionaries and models common neighbor information through distance encoding. However, since NAT learns **conditional representations** for each node’s neighbors (e.g., $u$’s neighbor $a$ and $v$’s neighbor $a$ will have different representations in $u$’s and $v$’s dictionaries), its complexity grows **quadratically** with the number of nodes. This necessitates neighborhood subsampling, where only a **limited subset of neighbors** is tracked for each node.
> >
> > In comparison, TNCN simplifies this process by maintaining a single (unconditional) representation for each node. It models pairwise information by aggregating the unconditional representations of common neighbors across multiple hops without imposing timestamp ordering restrictions. This gives TNCN linear complexity with respect to the number of nodes and eliminates the need for neighbor sampling. Additionally, TNCN can handle more complex edge interaction time orders, preserving more information about edge connectivity beyond just monotonic cases.
> >
> > For other concerns, please see the following comment.

---

> > > ### Author Response · Authors · 2024-11-21
> > >
> > > > W3. Theoretical Claims: The paper presents familiar results as novel contributions, such as the memory complexity for event aggregation, along with upper and lower bounds that are established in prior work (e.g., Caen, 1998). Presenting these as new theoretical results may be misleading.
> > >
> > > Thank you for your question regarding the theorems. We're sorry for any confusion, but we are **not** presenting an already existing conclusion in Caen (1998). In their paper, Caen (1998) proved a generic conclusion about the bound of the sum square of node degrees on general graphs. This result does not directly imply Theorem 5.6 (time complexity of dynamic graph link prediction methods). Instead, we are only leveraging this classic graph theory result as part of our proof to the temporal event complexity, where we first establish the connection between learning methods' time complexity and number of events aggregated, which is proved to be of the same order to the sum of squared node degrees. Only then does Caen's result apply. Further, the proof to k=2 case is also not trivial.
> > >
> > > Note that we clearly cited Caen (1998) in the proof provided in Appendix H. We have now added this citation in the main text too in our revision to ensure transparency and avoid potential misunderstandings.
> > >
> > >
> > > > W4. Experimental results: While combining memory and neighbor-based methods is interesting, the observed performance gains are minimal, with certain results (e.g., in Table 5) showing limited improvement using unclear metrics (possibly AUC or AP).
> > >
> > > Thank you for your question regarding the experimental results.
> > >
> > > In the TGB official setting, our TNCN model achieved relative improvements of 2.4%, 8.5%, and 15.9% over the second-best model on the Coin, Comment, and Flight datasets, respectively. This margin is **not minimal** in these large-scale TGB datasets. Meanwhile, this improvement is not achieved by sacrificing scalability like previous pairwise models. Most previous methods with pairwise modeling can not even finish the experiments on these three datasets under the official time restriction (see Table 2 in our paper). In contrast, our TNCN can not only achieve high efficiency but also gain significant performance improvments, validating our strong experimental performance.
> > >
> > > Regarding Table 5 (Table 6 now in the revision), we use Average Precision (AP) as the evaluation metric. We have updated it in revision. These datasets are relatively small in scale, which limits their representativeness for broader comparisons. On the Wikipedia and Reddit datasets, performance among top models is nearly saturated, making it challenging to discern differences in model capability. However, our model shows a significant improvement on Mooc and performs comparably to DyGFormer and FreeDyG on Lastfm.
> > >
> > > #### Questions:
> > > > Q1. Could you clarify the main novelty of your memory-based module and its differentiation from TGN?
> > >
> > > Please refer to the response to W2, part 1.
> > >
> > > > Q2. What is the specific metric used in Table 5? Understanding whether it's AUC or AP is essential for interpreting the results.
> > >
> > > Please refer to the response to W4.

---

> > ### Comment · Reviewer_BFZ2 · 2024-11-22
> >
> > *"The methods in large-scale benchmark TGB, including DyRep, TGN, CAWN, TGAT, TCL, Edgebank, and Graphmixer, primarily still do not involve pairwise information to make the final prediction."*
> >
> > This statement is inaccurate. For instance, TCL explicitly uses cross-attention between the representations of two nodes to capture pairwise information (as illustrated in Figure 2 of TCL). Similarly, CAWN employs causal anonymous walks and matches co-occurrence patterns between pairs of nodes. Therefore, it is incorrect to generalize that pairwise information is absent in these methods. In fact, leveraging pairwise information is now a common practice in dynamic graph learning.
> >
> >  *"The major novelty of TNCN lies in combining the memory-based pipeline with the NCN method to leverage pairwise information in an efficient way. Despite its simplicity, ..."*
> >
> > I understand the claimed novelty is the combination of memory-based models with NCN. However, I believe Section 4.1 belongs in the related work section rather than being presented as a core contribution of the paper. The combination is more incremental than groundbreaking and should be framed as such.
> >
> > *"A second perspective, which we did not emphasize in the abstract, is how our model achieves greater efficiency than graph-based framework methods that already incorporate pairwise information" and "In comparison, TNCN simplifies this process by maintaining a single (unconditional) representation for each node. It models pairwise information by aggregating the unconditional representations of common neighbors across multiple hops without imposing timestamp ordering restrictions."*
> >
> > These are indeed significant strengths of the paper, as I noted in my initial review. However, the presentation is misleading in several areas, particularly the method sections (Sections 4 and 5) and the abstract. For example, the claim that common neighbors (CN) are overlooked in dynamic graphs is incorrect. Furthermore, various non-novel aspects of the method are overemphasized, diminishing the overall clarity and impact of the genuine contributions.
> >
> > *"We're sorry for any confusion, but we are not presenting an already existing conclusion in Caen (1998)."*
> >
> > Thank you for clarifying and properly citing the paper in your main text. However, the first part of Theorem 5.6, which discusses the complexity $O(∣E∣)$ for memory-based methods, corresponds to the well-known complexity of message-passing methods. This is not a novel result and does not belong in the method section of a new research paper.
> >
> > *"Regarding Table 5 (Table 6 now in the revision), we use Average Precision (AP) as the evaluation metric."*
> >
> > Thank you for the clarification.
> >
> >
> > **Overall Feedback**
> >
> > This paper has several strengths, particularly in its engineering of combining existing techniques and its improvements to CN computation. However, the presentation is problematic. The main sections overemphasize existing techniques (e.g., Section 4.1, Theorem 5.6) while underselling the genuinely positive aspects, such as the efficiency improvements and the engineering contributions.
> >
> > The introduction and abstract are misleading, particularly the claim that CN is overlooked in dynamic graph learning, which is demonstrably false. Additionally, presenting incremental results as novel (e.g., in the methods section) detracts from the paper's credibility.
> >
> > Moreover, for an engineering-focused paper, I find that the results in Table 6 remain comparatively low against existing works. While there are good results highlighted in the main paper, a submission with low novelty must at least present exceptionally strong experimental results to be competitive at this conference.
> >
> > While the paper has potential, I strongly recommend a thorough rewrite to better align the narrative with its actual contributions. As it stands, I maintain my recommendation for rejection.

---

> > > ### Author Response · Authors · 2024-11-23
> > >
> > > Thanks for your comment.
> > >
> > > > 1. “For instance, TCL explicitly… Similarly, CAWN employs…”
> > >
> > > However, TCL doesn’t utilize any explicit pair-wise feature in its modeling. In fact, it just uses a cross-attention mechanism over the node, time and depth embeddings without constructing any pair-wise features, thus lacking the capability of even capturing CN between a given node pair. As for CAWN, the encoding of anonymous walks can capture some co-occurrence information, which we have missed in the rebuttal earlier. Nevertheless, we have discussed this method as a graph-based method in related work, section 2.2 in the initial paper. And CAWN can calculate CN if its first-step sampling has covered all of the neighbors from the starting node. However, CAWN suffers from several issues especially for its high time consumption from its walk sampling, hindering its application for large-scale datasets.
> > >
> > > > 2. “However, I believe Section 4.1 belongs in the related work section…”
> > >
> > > In fact we don’t present section 4.1 as a core contribution. Here we adopt memory-based backbone to construct our TNCN model, which plays an important role in the efficiency of TNCN. Our pipeline is established on such backbone, hence it’s naturally required to illustrate the memory-based module to give a better understanding of our model. And as you can see, we only use a few short lines to make it clear.
> > >
> > > > 3. “However, the presentation is misleading in several areas…”
> > >
> > > In fact we haven’t claimed that common neighbors (CN) are overlooked in dynamic graphs. From the beginning we have been making a statement that “traditional methods usually overlook pairwise information”. And we have also listed those majority of methods in TGB baselines that lack the modeling of pairwise features, and discussed those graph-based methods in the related work. In section 4 we highlight our CN extractor and NCN predictor with detailed paragraphs while only giving the illustration of memory-based backbone a few lines. In section 5 our primary work is formulating the issue of dynamic graph representation learning. And we have all clarified the necessary citation, without the intention to mislead readers of the novelty.
> > >
> > > > 4. “discusses the complexity O(∣E∣) for memory-based method…”
> > >
> > > The complexity result of memory-based network and message-passing methods are actually in the same format. However, the exact processes behind the two methods are quite different. Message-passing methods usually take the graph into a GNN, considering each edges of the static graph only once. As for memory-based backbone, it considers all of the historical interaction events, which happens to be equal to the total number of the events in dynamic graph. So the two conclusion are distinct even if they have the same numerical result.

---

> > > > ### Author Response · Authors · 2024-11-28
> > > >
> > > > Thanks for your comments about our representation about the “traditional methods overlook…” and about “CN”. While we have no intention to mislead the readers, we have taken your advice to modify the abstract and introduction parts to make it more clarified in the newly revised PDF. We have changed the statement and say that naïve memory-based and some attention-based methods neglect the pairwise feature while later methods have attempted to consider such information. However, they have less proportion in the real-world datasets like large-scale TGB, and may also suffer from high time consumption.
> > > >
> > > > As for table 6, the performances have reached saturation on Reddit and Wikipedia datasets. Many methods including memory-based and graph-based ones have achieved over 0.98 in AP. Hence these two datasets can not distinguish the capabilities of different models perfectly. For Mooc, our TNCN is the only method that achieving an AP over 90% under both transductive and inductive settings. For Lastfm, TNCN has achieved the best under inductive setting, and the second under transductive setting, which has a tiny margin than the best one(only 0.0019 absolutely and 0.2% relatively). So our TNCN also has the best average performance on these two datasets that can better distinguish models’ capability, substantiating our TNCN’s effectivity.
> > > >
> > > > As the discussion period is ending soon, we wonder whether our rebuttal is sufficient to address your concerns. If you find our revision and rebuttal satisfactory, would you please consider raising your score?

---

### Official Review · Reviewer_2N9E · 2024-11-01

**Soundness:** 3
**Presentation:** 3
**Contribution:** 2
**Rating:** 6
**Confidence:** 3

**Summary:**

This work proposes TNCN, which is  a temporal version of NCN based on a memory-based backbone. Comprosing with three key parts: the memory module,the temporal CN extractor,,and the NCN-based prediction head, TNCN improves the performance in terms of efficiency and effectiveness. Comparing with a diverse set of baseline models, the experiments on five datasets demonstrate its outstanding performance.

**Strengths:**

1. The experiments is substantial and the result is good.  Comparing with 9 baseline models, TNCN performs best on three of the five selected datasets, which emphasizes its effectiveness.
2. The method section is clearly described using formulas. With clear definition and detailed formulas, the method is well-presented.
3. There are proofs on the theorems in appendix, which improves the professionalism of the paper.

**Weaknesses:**

1. Since the process is relatively complicated, it is recommended to provide a pseudo code to make it easier for readers to understand.
2. I suggest that the experimental part be supplemented with an analysis of the hyperparameters, which can make the values of the hyperparameters more reasonable.

**Questions:**

1. High surprings values will degrade TNCN's performance. So will the model performance decrease monotonically as the surprise value increases? In other words, will TNCN have the best performance when the surprise value is the lowest?
2. Why is the result of TNCN-0∼2-hop-CN lower than TNCN-0∼1-hop-CN on the commen dataset in Table 2, which is different from the other three datasets?

---

> ### Author Response · Authors · 2024-11-21
>
> Thank you for acknowledging the performance and efficiency advantages of our model. We address your concerns as follows.
>
> > W1. Since the process is relatively complicated, it is recommended to provide a pseudo code to make it easier for readers to understand.
>
> Thank you for this suggestion. We have included pseudo code to illustrate the pipeline of TNCN, and it is now placed in Appendix E in the revised PDF.
>
> > W2. I suggest that the experimental part be supplemented with an analysis of the hyperparameters, which can make the values of the hyperparameters more reasonable.
>
> As per you request, we conduct experiments for the analysis of the hyperparameters to make their values more clear and reasonable. We particularly focus on num_neighbors and emb_dim&mem_dim, which have a larger influence on the performance of the model. The experimental results are shown as follows:
> | Wiki/num_neighbors | 10 | 12 | 15 | 18 | 20 |
> |---|---|---|---|---|---|
> | val mrr | 0.7433  | 0.7408  | 0.7412  | 0.7362  | 0.7418  |
> | test mrr | 0.7244  | 0.7193  | 0.7240  | 0.7187  | 0.7183  |
> | training time(s) | 26.35  | 27.17  | 29.49  | 30.32  | 31.56  |
> | test time(s) | 378.88  | 396.75  | 407.43  | 413.83  | 420.22  |
>
> | Coin/num_neighbors | 5 | 8 | 10 | 12 | 15 |
> |---|---|---|---|---|---|
> | val mrr | 0.7492  | 0.7378  | 0.7430  | 0.7450  | 0.7406  |
> | test mrr | 0.7687  | 0.7619  | 0.7701  | 0.7662  | 0.7601  |
> | training time(s) | 5936.19  | 6129.33  | 6406.00  | 6911.00  | 7529.01  |
> | test time(s) | 56605.45  | 57117.99  | 57292.00  | 57745.00  | 57925.00  |
>
> | Wiki/emb_dim&mem_dim  | 100 | 150 | 184 | 256 | 512 |
> |---|---|---|---|---|---|
> | val mrr | 0.7402  | 0.7373  | 0.7412  | 0.7426  | 0.7404  |
> | test mrr | 0.7178  | 0.7159  | 0.7240  | 0.7224  | 0.7235  |
> | training time(s) | 22.34  | 25.47  | 29.49  | 32.81  | 34.95  |
> | test time(s) | 378.28  | 399.77  | 407.43  | 420.45  | 459.86  |
>
> | Coin/emb_dim&mem_dim | 30 | 50 | 100 | 150 | 184 |
> |---|---|---|---|---|---|
> | val mrr | 0.7387  | 0.7405  | 0.7430  | 0.7436  | 0.7518  |
> | test mrr | 0.7591  | 0.7606  | 0.7701  | 0.7646  | 0.7699  |
> | training time(s) | 6228  | 6340  | 6406  | 6617  | 6721  |
> | test time(s) | 56694  | 57179  | 57292  | 58103  | 59411  |
>
> We have also attached these parameter analysis in the appendix C in our revised PDF, and we will make a better organization later.
>
> #### Questions:
> > Q1. High surprising values will degrade TNCN's performance. So will the model performance decrease monotonically as the surprise value increases? In other words, will TNCN have the best performance when the surprise value is the lowest?
>
> Thank you for highlighting this point. The “surprising value” refers to the ratio of test edges that were not observed during training. Datasets with a higher surprising value are generally considered to require more advanced inductive capabilities.
>
> We conduct experiments on the Wiki and Review datasets, calculating the MRR over ''surprising'' edges and ''unsurprising'' edges respectively. The result shown below reflects that the model can achieve a higher MRR on unsurprising edges than the surprising ones on both the datasets.
>
> | Dataset | Wiki | Review |
> |---|---|---|
> | MRR on surprising edge | 0.5299  | 0.4246  |
> | MRR on unsurprising edge | 0.8485  | 0.5065  |
>
> > Q2. Why is the result of TNCN-0∼2-hop-CN lower than TNCN-0∼1-hop-CN on the comment dataset in Table 2, which is different from the other three datasets?
>
> Neural Common Neighbors at different hops may contain varying information or even noise, thus having no guarantee to improve the model capability with the increase of the CN hops. We also conduct some experiments on Comment dataset and find that the performances are distinctive when using different hops of CNs. We can get some observations from the results:
> 1. The result of TNCN with all 1 hop CNs is higher than TGN, while TNCN with all 2 hop CNs is lower. This indicates that in this dataset, 1 hop CNs can benifit the capability of the model while 2 hop CNs will confuse it.
> 2. The result of TNCN with (0+1) hops CNs is higher than the (0+1+2), also suggesting the negative influence of 2 hop CNs.
> 3. (0+1) is higher than (1), and (0+1+2) higher than (1+2), facilitating the usefulness of introducing the 0 hop CNs.
>
> | Comment/hops | None (TGN)|1 | 0+1 | 0+1+2 | 1+2 | 2 |
> |---|---|---|---|---|---|---|
> | val mrr |0.636 |0.661  | 0.672  | 0.668  | 0.654  | 0.598  |
> | test mrr |0.622 |0.641  | 0.727  | 0.662  | 0.637  | 0.582  |

---

> > ### Comment · Reviewer_2N9E · 2024-11-25
> >
> > Thanks for the response. I have no further questions and I would like to keep my positive scores.

---

> > > ### Author Response · Authors · 2024-11-25
> > >
> > > Thank you for your positive feedback! We are delighted to your acknowledgement about our contribution. If you want to engage in any further discussion, please do not hesitate to reach out.

---

### Author Response · Authors · 2024-11-21
**Global Response**

We have revised our paper in the following aspects, which are marked in blue fonts:
1. Clarify the notation and transformation from node memory to embedding in section 4;
2. Add some parameter analysis in  Appendix C;
3. Attach pseudocode in Appendix E;
4. Add realistic case studies in Appendix G;
5. Add a comparison table between TNCN and NCN in section 4;
6. Complement contemporary work in related work;
7. Slightly modify some parts in abstract and description of Theorem 5.6.

We further address each reviewer's comments below.

---

### Comment · Area_Chair_KhKd · 2024-11-25
**Please reply to the authors' response.**

Dear reviewers,

The ICLR author discussion phase is ending soon. Could you please review the authors' responses and take the necessary actions? Feel free to ask additional questions during the discussion. If the authors address your concerns, kindly acknowledge their response and update your assessment as appropriate.


Best,
AC

---

### Meta-Review · Area_Chair_KhKd · 2024-12-18

**Metareview:**

The paper proposes TNCN, a temporal version of Neural Common Neighbor (NCN) for link prediction in temporal graphs. The paper demonstrates significant improvements in performance and efficiency, with TNCN outperforming popular GNN baselines and showing excellent scalability on large datasets.  However, some reviewers found the presentation misleading regarding the novelty of the method, and the core components of TNCN were seen as incremental rather than groundbreaking. Additionally, the performance gains were minimal on certain datasets, and there were concerns about presenting well-known results as novel contributions.

Despite these efforts, the concerns raised by some reviewer, particularly regarding the limited novelty and the misleading presentation, remain valid. Therefore, I recommend a reject for this paper.

**Additional Comments On Reviewer Discussion:**

The authors responded by clarifying the novelty of their approach, emphasizing the combination of memory-based models with NCN and the efficiency improvements. They revised the abstract and introduction to better highlight their contributions and addressed specific reviewer concerns about the presentation and theoretical claims.  They also provided further parameter analysis and case studies to support their claims.

---

### Decision · Program_Chairs · 2025-01-22

Reject